# Maternal dominance contributes to subgenome differentiation in allopolyploid fishes

Min-Rui-Xuan Xu [1,17], Zhen-Yang Liao [2,17], Jordan R. Brock [3,17], Kang Du [4,17], Guo-Yin Li[5,17], Zhi-Qiang Chen[6,17], Ying-Hao Wang[2], Zhong-Nan Gao[1], Gaurav Agarwal[7], Kevin H-C Wei [8,9], Feng Shao [10], Shuai Pang[6], Adrian E. Platts[3], Jozefien van de Velde [11], Hong-Min Lin[1], Scott J. Teresi [3], Kevin Bird [3], Chad E. Niederhuth[7], Jin-Gen Xu[12], Guo-Hua Yu[1], Jian-Yuan Yang[1], Si-Fa Dai[1], Andrew Nelson [13], Ingo Braasch [14], Xiao-Gu Zhang [1] ✉, Manfred Schartl [4,15] ✉, Patrick P. Edger [3,18] ✉, Min-Jin Han [16] ✉ & Hua-Hao Zhang [1,18] ✉

Teleost fishes, which are the largest and most diverse group of living vertebrates, have a rich history of ancient and recent polyploidy. Previous studies of allotetraploid common carp and goldfish (cyprinids) reported a dominant subgenome, which is more expressed and exhibits biased gene retention. However, the underlying mechanisms contributing to observed 'subgenome dominance' remains poorly understood. Here we report high-quality genomes of twenty-one cyprinids to investigate the origin and subsequent subgenome evolution patterns following three independent allopolyploidy events. We identify the closest extant relatives of the diploid progenitor species, investigate genetic and epigenetic differences among subgenomes, and conclude that observed subgenome dominance patterns are likely due to a combination of maternal dominance and transposable element densities in each polyploid. These findings provide an important foundation to understanding subgenome dominance patterns observed in teleost fishes, and ultimately the role of polyploidy in contributing to evolutionary innovations.

Whole genome duplications (WGD), also known as polyploidy, are an important recurrent process over evolutionary time that have contributed to the origin of novel phenotypes and driven species diversification across eukaryotes[1,2]. Polyploids are species that contain three or more complete sets of chromosomes in each nucleus, ranging from triploid (3 sets) to dodecaploid (12 sets)[3]. For example, two rounds of whole genome duplication, termed 1R and 2R events, are unique to vertebrates[4]. 1R preceded the origin of crown vertebrates, while 2R occurred in the lineage leading to bony vertebrates after the divergence of the cyclostome lineage[4–8]. Many retained duplicated genes from these two ancient polyploidy events have functionally diverged and are associated with the evolution of several novel structures including the neural crest, cartilage, bones and/or adipose tissue[9,10]. Similar patterns have also been reported following ancient polyploidy events in yeast[11] and plants[12]. Polyploids often evolve novel phenotypes and show greater phenotypic plasticity[13], which may explain certain polyploid lineages surviving mass-extinction events and exhibiting subsequent shifts in net diversification rates[14–16].

There are two main categories of polyploids; autopolyploids and allopolyploids[17]. Autopolyploids are formed from genome doubling

A full list of affiliations appears at the end of the paper. ✉e-mail: zcj7820@163.com; phch1@biozentrum.uni-wuerzburg.de; pedger@gmail.com; minjinhan@126.com; zhanghuahao_0824@126.com

involving a single diploid progenitor species, while the formation of allopolyploids involves genome doubling after hybridization of two or more diploid progenitor species[17]. Newly formed allopolyploid genomes may experience instability, as the previously separate genomes of each diploid progenitor species, known as subgenomes, have evolved independently and now coexist in a single nucleus[18,19]. The disruption in stoichiometry of highly dosage-sensitive components of macromolecular complexes and pathways, across regulatory, signaling and metabolic networks, can negatively affect fitness or be lethal[20]. Thus, partial to complete dominance of one subgenome over the other subgenome(s) may help resolve genetic incompatibilities[21]. Previous studies of ancient allopolyploids revealed that one subgenome may be dominantly expressed and over millions of years retain a significantly greater number of genes[22]. Subgenome dominance has been observed in many allopolyploids[23], to varying amounts, but not in all allopolyploids nor in any autopolyploids (e.g. allopolyploid cereal grass Teff[24]). Thus, the underlying genetic and/or epigenetic mechanisms driving expression dominance remains poorly understood[19]. Previous studies have shown that densities of transposable elements near genes are predictive of which subgenome is more highly expressed (i.e. dominant)[25]. However, if and how much genetic divergence of the diploid progenitors contributes to subgenome expression dominance has yet been evaluated in allopolyploids and especially in vertebrates.

An additional whole genome duplication, termed TGD or 3 R, occurred in the teleosts fish lineage, estimated 225–350 million years ago, at the base of the largest and most diverse group of vertebrates (>30,000 extant species)[26]. Some clades including Salmonidae (Teleostei: Salmoniformes)[27,28], Cyprinidae (Teleostei: Cypriniformes)[29,30] and Corydoradinae[31] have undergone their own, independent fourth rounds (4Rs) of polyploidization[32,33]. Cyprinids, the carp family, contain roughly 600 polyploid species derived from potentially at least thirteen polyploidization events[34]. The family is delineated into eleven subfamilies, including Cyprininae that consists of eleven tribes, of which seven (Schizopygopsini, Cyprinini, Torini, Probarbini, Spinibarbini, Barbini, and Schizothoracini) are largely composed of polyploids[35], Thus, cyprinids are an ideal model family for investigating subgenome evolution following multiple independent polyploid events within vertebrates.

To date, to the best of our knowledge, subgenome-resolved assemblies of only three allopolyploid species from the Cyprinini tribe are publicly available, including the common carp (*Cyprinus carpio*)[29,36], goldfish (*Carassius auratus*) ($2n = 4x = 100$)[37–39], and the hexaploid Prussian carp (*Carassius gibelio*) ($2n = 6x = 150$)[30,40]. Some evidence for subgenome expression dominance was uncovered from the analysis of both the common carp and goldfish genomes[29,36,37]. However, no evidence for subgenome dominance at the transcriptome level was observed following the analysis of the hexaploid Prussian carp genome[30]. Comparative genomic analysis of the Prussian carp revealed biased duplicate gene retention of certain genes towards one subgenome[30]. This suggests that the genomes of cyprinine allopolyploid cyprinid fishes may exhibit subgenome dominance to varying levels. In this context, the role of transposable element differences, parental effects and/or genetic divergence of diploid progenitor species contributing to observed subgenome expression dominance remains poorly understood. Therefore, the evaluation of multiple independently derived cyprinine allopolyploids can provide valuable new insights into the underlying mechanisms of subgenome dominance.

A robust phylogenomic framework for the subfamily Cyprininae is needed to phylogenetically localize polyploidy events and investigate the underlying genetic mechanisms contributing to subgenome dominance in allopolyploid fishes. However, the maternal and paternal diploid progenitors of known polyploids in this group remain largely unknown. A recent study[34] tried to address this point within this group using three single-copy nuclear loci, but the phylogenetic history of these three genes may not reflect the true history of species relationships within this subfamily. Phylogenomic analyses based on hundreds of orthologous markers from across the genome should reflect a more accurate evolutionary history of the species and more likely to reveal the diploid progenitors of allopolyploids[41].

In the present study, we thus aim to resolve the phylogenetic relationships among several key Cyprininae species, uncover the polyploid origin of three allopolyploid species, identify the closest extant relatives of their diploid progenitors and investigate subgenome dominance and its genetic basis in the allopolyploids. To accomplish these goals, we assemble de novo high-quality reference genomes of twenty-one cyprinid fishes from across five subfamilies using PacBio HiFi long reads. Furthermore, we generate transcriptome data from several distinct organs to investigate subgenome expression dominance in three allotetraploids. Our study provides new insights into the evolutionary history of Cyprininae, including the identification of maternal and paternal diploid progenitor lineages of three independently formed allopolyploids, the genetic basis of subgenome dominance in these allopolyploids, and new large-scale genomic resources for the community as a foundation for future studies.

## Results
### Genome assembly and annotation
Whole genomes of 21 cyprinid fishes were sequenced with PacBio CCS (circular consensus sequencing) reads with an average of 32.34-fold coverage and Illumina paired-end 150 bp reads with an average 66.86-fold coverage, in total yielding 2.24 trillion base pairs (Tbp) of raw read data (Fig. 1; Supplementary Tables 1 and 2). These datasets were de novo assembled using Hifiasm[42], yielding high-quality genomes with an average contig N50 size of 23 Mb (Supplementary Table 3). The new assemblies ranged in size from 0.81 to 1.83 Gbp, similar to the estimated genome sizes obtained from *k*-mer analysis of Illumina reads (Supplementary Fig. 1 and Table 4). A high percentage (>99%) of Illumina reads aligned against the assembled contigs and high BUSCO (Benchmarking Universal Single-Copy Orthologs)[43] scores (average 95.60%, from 91.7 to 96.6%), suggesting that the biggest proportion of the genomes was assembled (Supplementary Data 1 and Table 5).

Previous phylogenetic work using three single-copy nuclear loci suggested that three species *Procypris rabaudi* (Tribe Cyprinini), *Spinibarbus sinensis* (Tribe Spinibarbini) and *Luciobarbus capito* (Tribe Barbini) are likely tetraploids ($2n = 4x = 100$)[44–46]. To generate chromosome-level genomes, high-throughput chromosome conformation capture (Hi-C) reads, at ~100-fold coverage per haplotype, were obtained and scaffolded for each tetraploid with the ALLHiC algorithm[47] (Supplementary Tables 6–8). In total, 94.43%, 97.56% and 98.83% of all bases corresponding to *S. sinensis*, *P. rabaudi* and *L. capito* genomes were assigned to 50 pseudo-molecules (chromosomes) after manual curation (Supplementary Tables 9–11). Strong contact signals of the Hi-C data for all chromosomes of each genome suggest high quality of chromosome-level scaffolding (Fig. 2a, Supplementary Figs. 2a and 3a, 4–6).

Homology-based and RNA sequence-based gene predictions were used to annotate all genomes after masking transposable elements (TEs), simple sequence repeats (SSRs), and tandem repeats. The final annotated gene numbers for the three allopolyploids, *P. rabaudi*, *L. capito* and *S. sinensis*, were 45,857, 43,211 and 49,999 (Supplementary Data 2), respectively, which were comparable to those of two famous cyprinid fishes common carp (*Cyprinus carpio*) (47,924) and goldfish (*Carassius auratus*) (48,857)[29]. The gene number of the rest eighteen species ranged from 23,658 to 32,381, which are similar to the 24,770 for *Onychostoma macrolepis*[48] and 27,263 for grass carp (*Ctenopharyngodon idella*)[49]. BUSCO analysis[43] was conducted to evaluate the completeness of these annotations, which contain an average of 91.6% (from 87.4 to 96.7%) complete BUSCO gene sets (Supplementary Table 12).

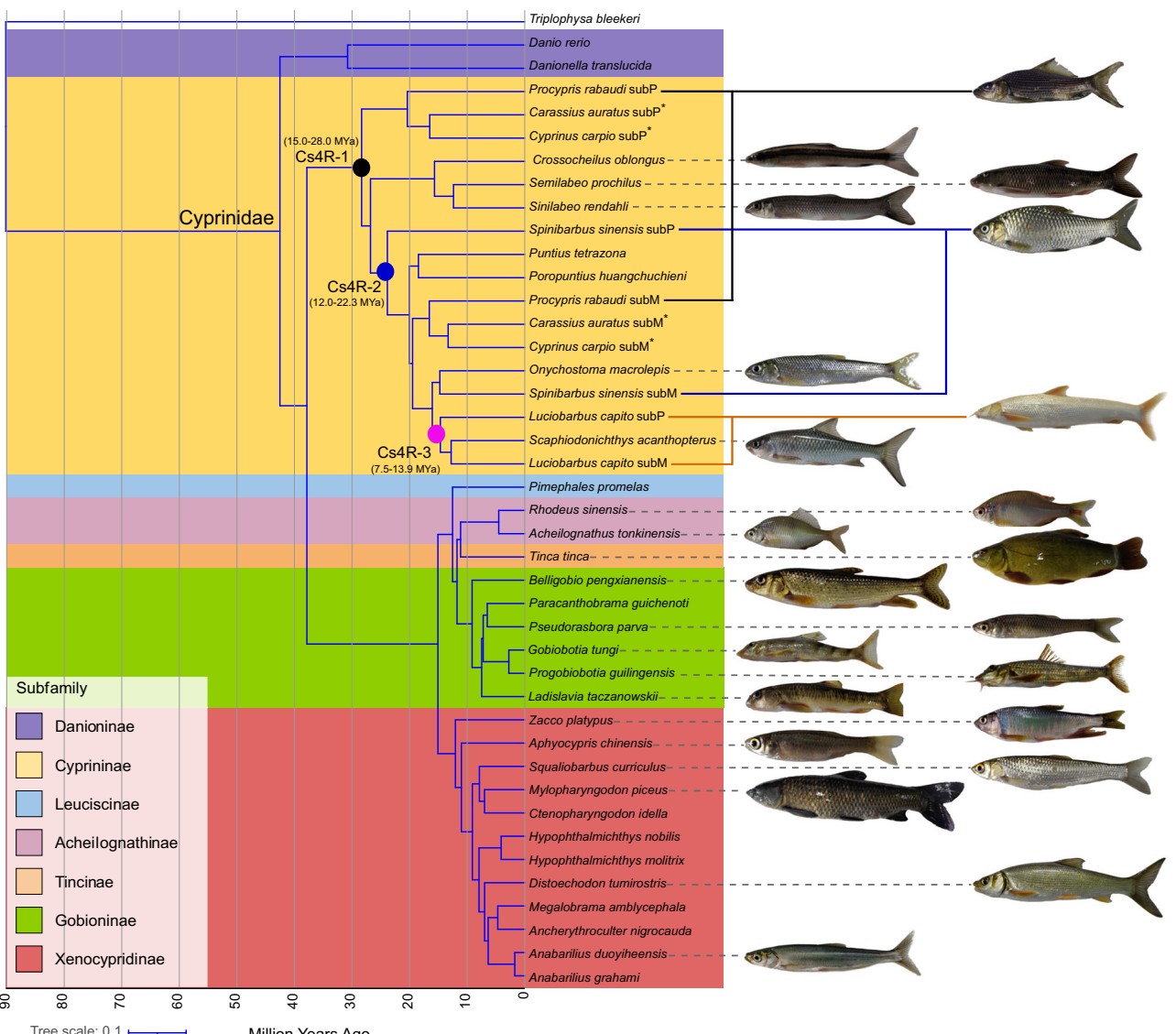

**Fig. 1 | Phylogeny and divergence time of fishes within the Cyprinidae family.** Species tree constructed using IQ-TREE based on CDS of 300 one-to-one orthologues from 37 studied species was shown in Supplementary Fig. 13. *Triplophysa bleekeri* was used as the outgroup. Divergence time between all fishes or subgenomes of five allotetraploids was inferred by MCMCTree (Supplementary Fig. 12). All the fish images were created in this study. Black, blue and yellow solid circles represent the divergence timepoints of the diploid progenitor lineages for the three independent cyprinid-specific whole genome duplication (Cs4R). However, they do not represent the timepoints of the three WGDs. The numbers in parentheses represent the time of WGD. Fishes belonging to the corresponding subfamilies of the Cyprinidae family were shown using unique background color.

The overall TE content in the 21 sequenced species ranged from 40.87% to 59.18% (Supplementary Table 13). Predicted genome size was positively correlated with TE content (Supplementary Fig. 7) (Spearman's rank correlation coefficient: 0.9, *p* value = 3.3e-9). The most abundant repeat class of all species was DNA transposons (from 17.03 to 37.81%), of which TC1/mariner, hAT, and CMC were the three top enriched superfamilies (Supplementary Data 3). Long terminal repeats (LTRs) account for an average of 11.09% (from 7.11 to 15.26%) of the genomes, which is higher than reported for zebrafish (*Danio rerio*) (6.0%)[50]. Most of our sequenced fishes contained similar long interspersed nuclear element (LINE) content (average 4.03%; from 2.44 to 5.3%) with that of zebrafish (4.1%) but fewer short interspersed nuclear elements (SINEs) (average 0.82%; from 0.2 to 3.42%) than zebrafish (3.1%).We also observed that the median age of DNA transposon families in our sequenced genomes were typically older than those of both LTR and LINE families (Supplementary Fig. 8), which was also found in the zebrafish[50].

## Subgenome-resolved assemblies and allotetraploid origin of three fishes

In this study, we provide several additional lines of evidence to support that *P. rabaudi*, *L. capito* and *S. sinensis* are polyploids. First, more than 59% of BUSCO genes of their predicted protein-coding genes were duplicated (Supplementary Table 12). Second, the assembled genome sizes (from 1.6 to 1.8 Gb) of these species' genomes (Supplementary Table 3) were approximately double that of diploid species from the subfamily Cyprininae, including *O. macrolepis* (886.5 Mb)[48] and *Poropuntius huangchuchieni* (1 Gb)[51], and similar to that of two reported tetraploids, common carp (1.68 Gb) and goldfish (1.65 Gb)[29]. Third, the number of Hox clusters identified in their genomes was twice that of the zebrafish genome (Supplementary Data 4 and Fig. 9). Fourth, a 2:1 relationship in orthologous syntenic genes was observed between each tetraploid and *O. macrolepis* through synteny analysis (Fig. 2b and Supplementary Figs. 2b and 3b).

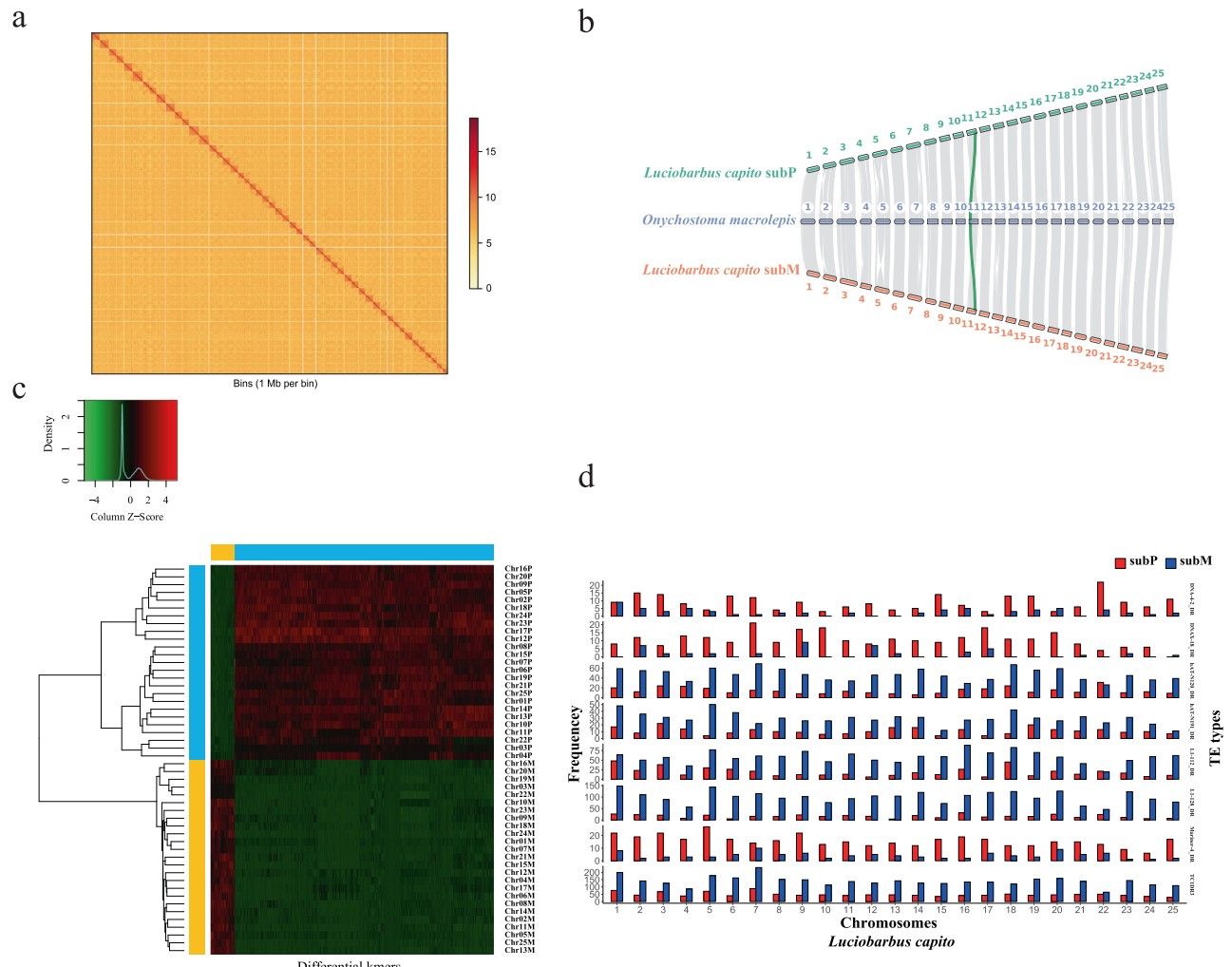

**Fig. 2 | Evidence for the allotetraploid origin of *L. capito*. a** Intensity signal heat map of the high-throughput chromatin conformation capture (Hi-C) chromosome interaction. **b** Syntenic relationships between *O. macrolepis* and *L. capito* subP and subM. The green band showed one example of a collinearity gene between homologous chromosomes. **c** Heatmap and clustering of differential k-mers. The x-axis, differential k-mers; y-axis, chromosomes. The vertical color bar, each chromosome is assigned to subP and subM; the horizontal color bar, each k-mer is specific to subP and subM. **d** TE frequency on chromosomes showing subP and subM biased distributions in the tetraploid genome of *L. capito*. Evidence supporting the allotetraploid origin of *S. sinensis* and *P. rabaudi* was present in Supplementary Figs. 2 and 3.

Multiple alignments of orthologous genes between each tetraploid and *O. macrolepis* successfully identified two subgenomes, each of which included 25 chromosomes (Fig. 2b and Supplementary Figs. 2b and 3b). To assign each chromosome to a subgenome, a method similar to SubPhaser[52], a novel subgenome-phasing algorithm using subgenome-specific k-mers as markers, was applied. The allopolyploid origin of several previously determined allopolyploid plants as well as the common carp and African clawed frog *Xenopus laevis* was supported using this strategy[52]. Therefore, the presence of repetitive *k*-mers, which are exclusively or highly enriched towards one subgenome, were sought for each of the three polyploids. We confirmed that two distinct subgenomes, termed 'subP' and 'subM' (see below for designation), of each tetraploid could be determined based on a suite of 15-mers with unique distribution patterns along each homoeologous chromosome pair, supporting an allotetraploid origin of these three species (Fig. 2c, Supplementary Figs. 2c, 3c and 10). To further verify the polyploid origin (i.e. auto- vs allo-polyploid), we adapted another strategy that involves analyzing TE (transposon) types and abundances that has been successfully employed to confirm the polyploid history of the African clawed frog[53], blueberry[54], sterlet sturgeon (*Acipenser ruthenus*)[55], the goldfish[39] and Prussian carp

(*Carassius gibelio*)[30]. This approach is based on the hypothesis that relics of unique transposon types and abundances specific to the two parental species can be used as markers to partition each chromosome to a particular subgenome in an allopolyploid. Frequency analyses of TEs identified between 8 and 16 transposon types in each polyploid genome that were enriched differentially in the subP and subM (Fig. 2d, Supplementary Figs. 2d, 3d and 11,Table 14). These results collectively support an allopolyploid origin for these three polyploid fishes.

## The divergence time and hybrid origin of allotetraploids

To estimate the divergence time of each subgenome, we established one-to-one ortholog gene sets from two putative diploid ancestors (*O. macrolepis* and *Scaphiodonichthys acanthopterus*) and the subP and subM genomes of three allotetraploids (*P. rabaudi*, *L. capito* and *S. sinensis*) and calculated the pairwise synonymous substitutions (*Ks*). The divergence-time of diploid progenitors (subgenomes), served as the upper bound estimate of the polyploid event, and can be deduced based on the *Ks* age distributions of the orthologous pairs (Fig. 3a). We found that the two subgenomes of *L. capito* diverged approximately 7.5 to 13.9 million years ago (Mya), which is the most recent date

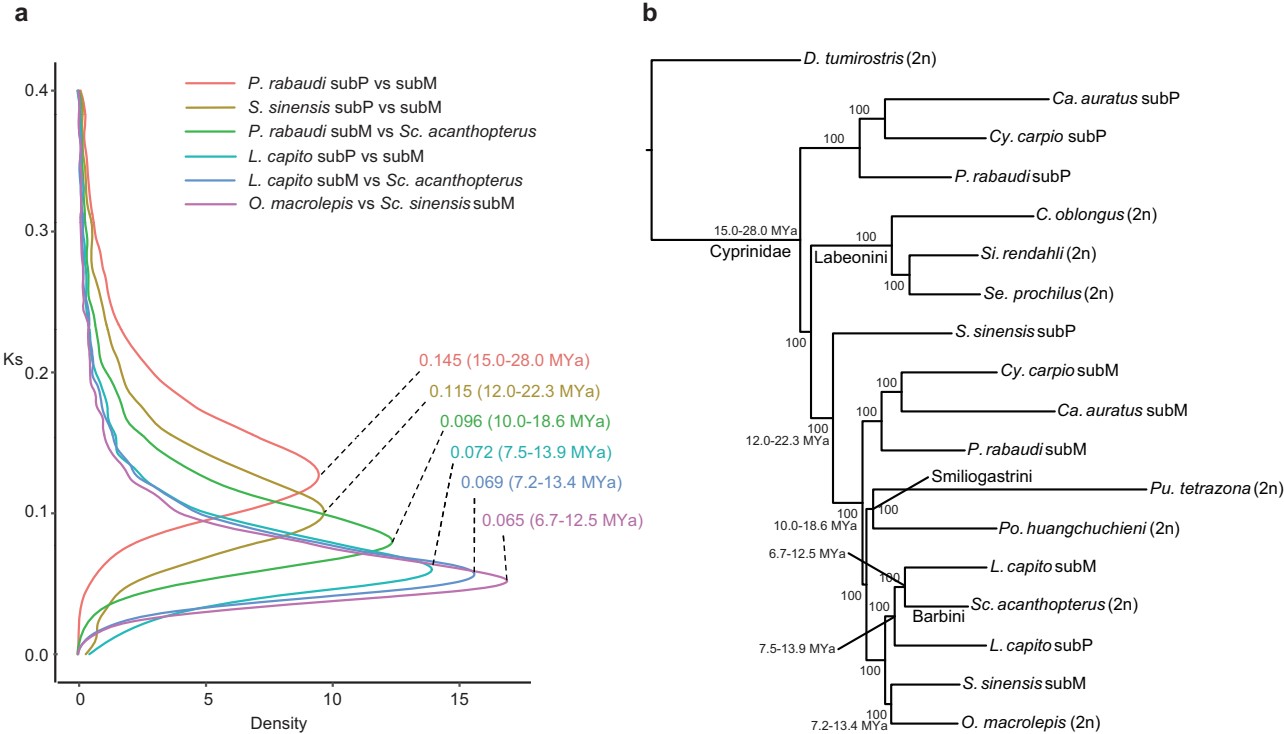

**Fig. 3 | Time estimates of polyploid events and phylogeny relationship between potential diploid ancestors and allotetraploids. a** Distribution of synonymous substitution rates (*Ks*) between species and between subP and subM. Numbers at distribution peaks indicate median *Ks* values. **b** The phylogenetic relationship of possible diploid ancestors and allotetraploid subgenomes. The ML tree based on CDS of 1669 one to one orthologs from 13 species was created by IQ-TREE. We used *Distoechodon tumirostris* as the outgroup.

estimate among the allopolyploids examined in this study (Fig. 3b). In comparison, the divergence of the *P. rabaudi* subgenomes is estimated at ~15 to 28 Mya. This estimate is similar to the previous divergence times estimates (13.5 to 25.6 Mya) of the subgenomes of common carp and goldfish[29]. The results from our phylogenetic analyses further confirmed that *P. rabaudi*, common carp and goldfish likely share a common polyploid event, with subP and subM of each species in monophyletic clades (Fig. 3b). Lastly, the divergence of the subgenomes of *S. sinensis* was estimated at 10 to 18.6 Mya (Fig. 3a). Therefore, these three allopolyploid cases, with varying divergence estimates among subgenomes (Figs. 1 and 3, Supplementary Fig. 12), provides a suitable framework to examine whether genetic divergence of the diploid progenitors contributes to subgenome expression dominance.

Mitochondrial genomes (mtDNA) are almost exclusively inherited from maternal progenitors[56], whereas nuclear protein-coding genes are biparentally inherited. Therefore, a comparison of the mtDNA phylogenetic tree and nuclear gene trees enables the identification of maternal and paternal diploid progenitors for allopolyploids[34]. Our phylogenetic analyses using *Triplophysa bleekeri* or zebrafish as an outgroup provide strongly supported estimates for species relationships and the monophyly of Cyprininae (Fig. 1 and Supplementary Figs. 13 and 14). Furthermore, these analyses revealed three independent polyploidization events: one shared by *P. rabaudi*, common carp, and goldfish (Cs4R-1), one in *S. sinensis* (Cs4R-2) and one in *L. capito* (Cs4R-3) (Figs. 1 and 3b), consistent with a previous study[34].

Based on the aforementioned phylogenetic analyses (Figs. 1 and 3b, Supplementary Figs. 13 and 14) and the mitochondrial tree (Supplementary Fig. 15), the subP and subM of these five species denotes the paternal and maternal subgenome, respectively. These analyses also supported three independent allopolyploid origins. The maternal subM of common carp, goldfish and *P. rabaudi* is most closely related to Tribe Barbini or Acrossocheilini, and the paternal subP is most

closely related to Tribe Labeonini. Similarly, a closely related species of Acrossocheilini could have served as the diploid progenitor of the *S. sinensis* subM, whereas its subP was the descendent of an ancestral fish much older than Smiliogastrini. The formation of *L. capito* was probably the result of hybridization of two diploid relatives from Barbini. To further confirm the above conclusion, phylogenetic analyses with the whole-genome alignment (WGA) of 13 species, the fourfold degenerate sites (4dtv) in 1669 genes and CDS of 1669 individual genes were performed. The topologies of all these trees were congruent with each other (Supplementary Figs. 16–19). Meanwhile, we also observed the differences between overall consensus species tree and individual gene trees (Supplementary Fig. 18), implying that these topological conflicts may be as a result of incomplete lineage sorting (ILS) and introgression.

**Evolutionary mechanisms of duplicated ohnologs**

Generally, there are four major evolutionary fates for duplicated genes (ohnologs) derived from polyploidy events, including 1. duplicate gene retention due to dosage-balance constraints or selection favoring increased dosage of gene products[11,20,57], 2. gene loss or pseudogenization of one duplicate copy[58,59], 3. subfunctionalization, the partitioning of ancestral gene functions among the two duplicate gene copies[60] and 4. neofunctionalization, the evolution of novel gene functions in one or both duplicate gene copies[4]. To investigate the frequency of each fate among ohnologs, we analyzed the expression levels across six tissues for a set of positionally conserved syntenic ohnologs (7,040 total) that were present in single copy in the genomes of two diploids (*O. macrolepis* and *Sc. acanthopterus*) and retained in duplicate in all three allotetraploid genomes. We identified 4884 (69.68%) to 5,345 (75.92%) gene pairs that had expression patterns consistent with duplicate retention due to dosage-selection, 226 (3.21%) to 348 (4.94%) due to non-functionalization, 9 (0.13%) to 14 (0.199) due to subfunctionalization, and 223 (3.17%) to 420 (5.96%) due

to neofunctionalization (Supplementary Table 15). Examples of expression divergence consistent with subfunctionalization and neofunctionalization for each allotetraploid are shown in Supplementary Fig. 20. However, we should notice that the low level of subfunctionalization inferred (<1% of gene pairs) could be due to the relatively small number of tissues examined. Our results were also consistent with those previously reported in goldfish[37], which suggests the most common mechanism for duplicate gene retention in these allopolyploid cyprinines since their 4R event is due to dosage-constraints. However, mechanisms for duplicate gene retention are not strictly inferable because a multilevel set of phenomena that range across WGD[61].

### Subgenome dominance patterns in three allotetraploids–Gene Fractionation (loss)

Allopolyploids face the unique challenge of integrating two subgenomes, which evolved independently in the diploid progenitors since their most recent common ancestor, that now reside in a single nucleus[3]. One way to resolve potential genetic or epigenetic conflicts is "subgenome dominance"[22], which results in one subgenome being dominant over the 'submissive' subgenome(s)[59]. The dominant subgenome not only has higher gene expression but also retains a greater number of ohnologs compared to the submissive subgenome(s). To better understand the dynamics among subgenomes of our three sequenced allotetraploids, we compared their gene loss (fractionation), gene expression level, the density of TEs near ohnologs, constraint on conserved noncoding sequences (CNSs), DNA methylation patterns and 3D genome structure.

To examine gene fractionation (loss) differences among subgenomes, gene retention patterns between the two subgenomes (paternal subP and maternal subM) of the three allopolyploid species (*P. rabaudi*, *S. sinensis* and *L. capito*) were examined relative to the diploid references from zebrafish, *O. macrolepis* and *Sc. acanthopterus*. These results revealed that in all cases, the maternal subM showed slightly higher gene retention rates relative to the paternal subP (Fig. 4a, Supplementary Figs. 21–29, Table 16). Compared to the reference zebrafish, subM showed 2.815% higher gene retention in *L. capito*, 0.427% higher gene retention in *P. rabaudi*, and 0.819% higher gene retention in *S. sinensis* relative to subP. However, these patterns are not supportive of strong subgenome dominance patterns as has been reported in some plant allopolyploids (e.g. Chinese cabbage[62]).

Ohnolog retention bias of certain sets of genes, including BUSCO genes towards one subgenome has been recently reported for the Prussian carp, goldfish and common carp[30]. Similarly, we found that the number of BUSCO singleton genes in maternal subM was significantly higher than those in subP for all three allotetraploids (Supplementary Fig. 30). For example, subM of *S. sinensis* has retained 609 complete and single copy BUSCO genes, compared to only 448 in subP ($X^2$ text; *p*-value < 0.001). Next, we performed GO (Gene Ontology) analysis of the genes that returned to single copy in subP and subM. Functional enrichment analysis revealed that similar GO term classes were identified for all species, including mitochondrial related processes, nc/rRNA processes and DNA repair (Supplementary Figs. 31–36). These GO terms were also identified as returning to singleton state post-WGD from a previous analysis of the Prussian carp, goldfish, and common carp genomes[30].

### Subgenome dominance patterns in three allotetraploids - Tandem Gene Duplications

A previous study investigating subgenome dominance in octoploid strawberry revealed that the dominant subgenome retained a significantly greater number of tandem duplicated genes[63]. Here, we uncovered a similar pattern for retained tandem gene duplications being biased towards the maternal subM in all three allotetraploid cyprinids (Supplementary Table 17). First, significantly more tandem duplicates are encoded on the maternal subM compared to the paternal subP (5283 vs 4579 in *L. capito*, 6042 vs 5268 in *S. sinensis*, 4929 vs 4564 in *P. rabaudi*) (each species $X^2$ test p-value < 0.001). Second, a greater number of tandem gene arrays were observed in the maternal subM compared to paternal subP (1915 vs 1803 arrays in *L capito*, 2155 vs 1992 arrays in *S sinensis*, 1826 vs 1800 in *P. rabaudi*). Lastly, the maternal subM genomes contained a greater number of larger tandem arrays (>5 tandem genes) than the paternal subP (94 vs 52 arrays in *L. capito*, 114 vs 82 arrays in *S. sinensis*, 79 vs 54 arrays in *P. rabaudi*). An analysis of protein family domains revealed an enrichment of functions associated with the immune system for retained tandem duplicates in these subgenomes (Fig. 5).

### Subgenome dominance patterns in three allotetraploids–Pangenome Analysis

We ran a pangenome analysis of 36 available cyprinid fish genomes to identify shared and unique orthogroups. Orthogroups shared by all 36 species were defined as core genes, those shared by 32–35 species were defined as softcore genes, those shared by 2–31 species were defined as dispensable genes, and those only presented in one species were defined as private orthogroups (Supplementary Fig. 37a). If a gene appears in either subP or subM, we considered that the allopolyploid species shared that gene. A total of 50,585 orthogroups were identified, including 25,325 (50.1%) dispensable, 11,858 (23.4%) private, 9,874 (19.5%) softcore, and 3501 (6.9%) core orthogroups (Supplementary Fig. 37a). For core genes that returned to single copy, a bias towards retention on the maternal subM was observed for both *L. capito* and *S. sinensis* ($X^2$ text; p-value ≤ 4.6e-4) (Supplementary Fig. 37b). Similarly, we observed a pattern of subM bias for core gene retention in the goldfish and common carp genome ($X^2$ text; *p*-value ≤ 7.2e-6) (Supplementary Fig. 37b).

### Subgenome dominance patterns in three allotetraploids–Gene Expression

Next, we examined global gene expression patterns of each subgenome for the three focal allopolyploids. Gene expression abundance analysis across six tissue types revealed broadly consistent trends of slightly higher median expression of genes in maternal subM relative to paternal subP (Supplementary Fig. 38). Furthermore, more significantly biased expressed ohnologs towards the subM were identified when analyzing retained duplicates present in both subgenomes (Fig. 4b, Supplementary Fig. 39). These results suggest that more genes exhibit expression bias toward the maternal subM in five of six, six of six, and five of six tissues in *S. sinensis*, *L. capito*, and *P. rabaudi* respectively (Supplementary Table 18). This pattern is largely retained for those 7040 positionally conserved (syntenic) ohnologs maintained in a 1:1:2:2:2:2 ratio among diploids and tetraploids with six, three, and three tissues being significantly biased toward the subM in *S. sinensis*, *L. capito*, and *P. rabaudi* respectively (Supplementary Table 19).

Proximal TE loads may be linked to expression difference among genes[64], which has been shown to be associated with observed subgenome expression dominance in certain allopolyploid plants[25,65]. To test this hypothesis, the density of TEs near genes was evaluated in the three allopolyploids. TE density up- and down-stream of genes in *S. sinensis*, *L. capito* and *P. rabaudi* displayed no general bias globally in subP or subM (Fig. 4c, Supplementary Fig. 40). For those genes that were found to exhibit expression bias in at least three tissue types toward subP, we observed only *S. sinensis* to have significantly higher TE density upstream of genes in subM (Supplementary Fig. 41). Those genes with biased expression towards the maternal subM showed significantly lower TE density upstream and downstream of genes in subM for *L. capito* and upstream of genes in subM for *P. rabaudi* (Supplementary Figs. 42).

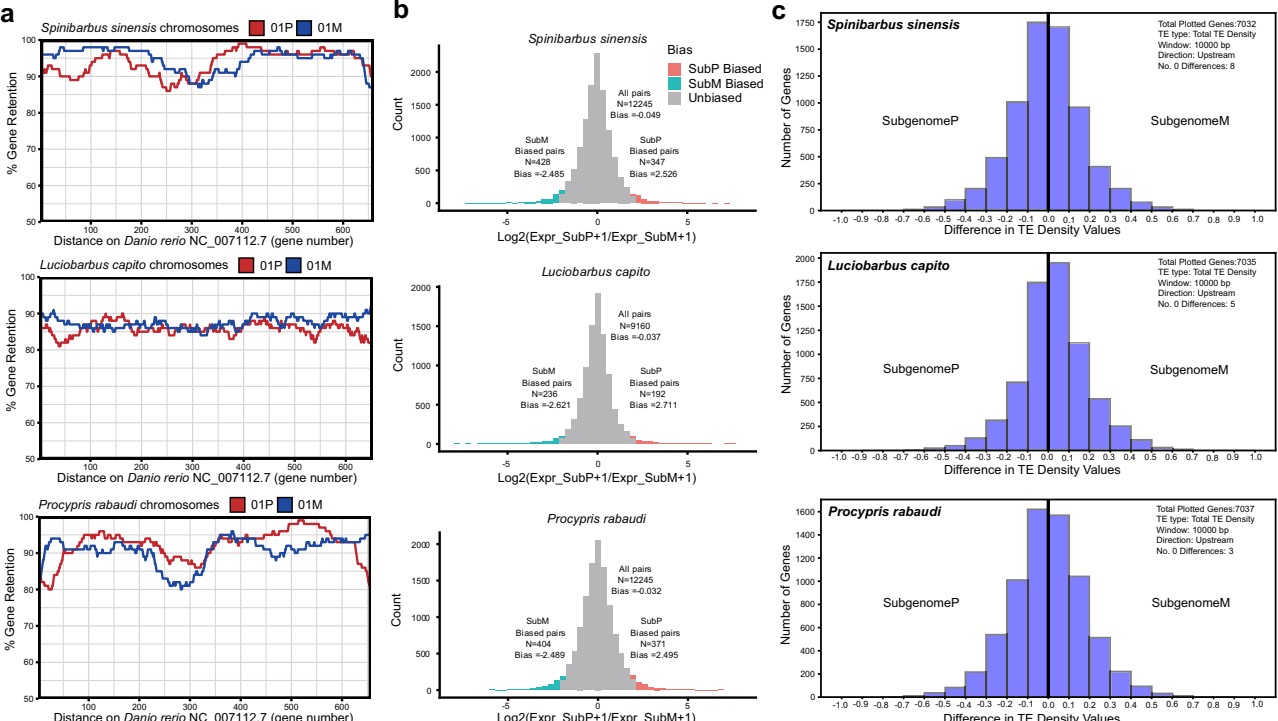

**Fig. 4 | Gene fractionation, gene expression and TE density of subgenomes.**
**a** Subgenome fractionation of allotetraploids *S. sinensis*, *L. capito*, and *P. rabaudi* relative to the diploid *Danio rerio*. Gene retention in focal tetraploid subP (red) and subM (blue) was calculated in 100 gene sliding windows and displayed for chromosome 1 of each tetraploid species. Gene retention of the rest chromosomes of each tetraploid was showed in Supplementary Figs. 18–20; **b** Global subgenome expression bias in the brain tissue of studied tetraploid species, with biased gene counts colored according to subP (red) and subM (blue). Subgenome expression bias in the rest five tissues eye, gill, heart, liver and muscle of studied tetraploid species was shown in Supplementary Fig. 36; **c** Histograms of differences in TE density values of subP and subM syntelogs of *S. sinensis*, *L. capito*, and *P. rabaudi*. Density values were calculated for all TEs in a 10,000 bp window upstream of genes and difference values were calculated by subtracting TE density of subM syntelogs from subP syntelogs. Negative values represent higher TE density for subM syntelogs, whereas positive values reflect higher TE density for subP syntelogs.

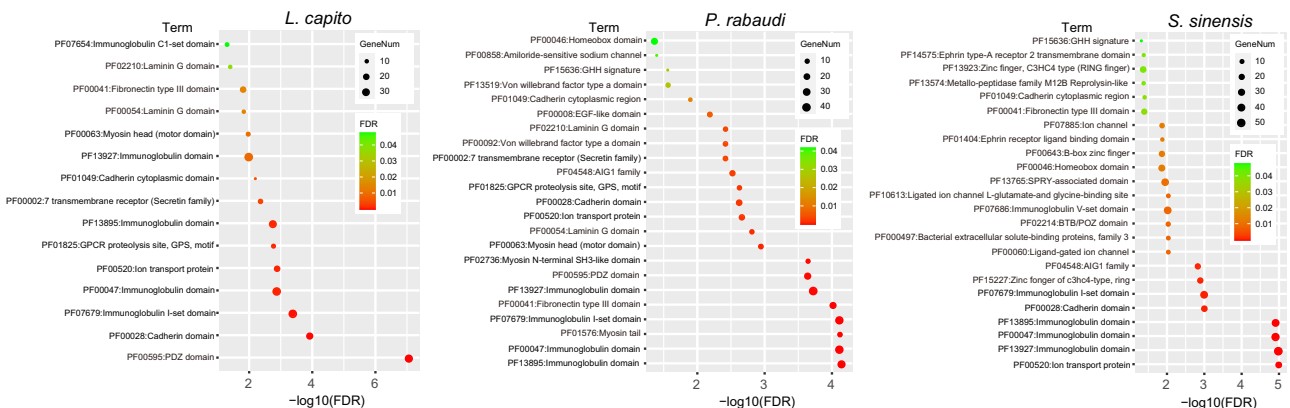

**Fig. 5 | Functional enrichment analysis of tandem duplicate genes in three allotetraploid genomes.** PFAM domain analysis shows an enrichment of tandem duplicate genes associated with the immune system, including 'PF07679 Immunoglobulin I-set domain' of *L. capito* (FDR *p*-value = 0.00037), *P. rabaudi* (FDR *p*-value = 7.37e-05),and *S. sinensis* (FDR *p*-value = 0.00093).

## Subgenome dominance patterns in three allotetraploids–Conserved Noncoding Sequences

Previous studies reported that variation in conserved noncoding sequences (CNSs), which contain cis-regulatory sequences, was related to the level of gene expression, such that the loss of CNSs leads to the absence or divergence of gene expression[66,67]. Thus, we sought to compare selective constraints of CNSs across subgenomes of each of the three allopolyploids. In total, we identified 864,476 CNSs spanning a total 44.0MBase, mean CNS length of 51nt, 4.5% of assembled genome

lengths, which is similar to the 3%-8% values described elsewhere for vertebrates)[68]. Overall constraint on CNSs was found to be marginally higher for subP in *P. rabaudi* and *S. sinensis*, whereas constraint was marginally higher for the subM in *L. capito* (Supplementary Table 20).

## Subgenome dominance patterns in three allotetraploids–DNA Methylation

We also tested the hypothesis that DNA methylation patterns in genes and TEs in the extant relatives of diploid progenitor species,

and thus subgenomes within allotetraploids, may explain observed subgenome expression bias patterns[19,25,64]. Whole-genome bisulfite sequencing of the muscle tissue from two diploid ancestors and three allotetraploids was performed (Supplementary Table 21). Levels of CH (where H = C, A, T) methylation were very low (consistently <0.5%) in genes of all five species (Supplementary Fig. 43), which was also observed in the common carp genome[36] and is typical of somatic tissues in humans[69]. Therefore, we focused on CG methylation for all subsequent analyses. A similar pattern of CG methylation was observed within the gene body and 2 kb flanking regions in case of the diploid *Sc. acanthopterus* and the three allotetraploid species (Supplementary Fig. 44). However, for *O. macrolepis*, there was much lower CG methylation levels ~1 kb upstream up to the transcriptional start site (TSS) and higher levels throughout the gene body and ~1 kb downstream. No difference in CG methylation was observed among the diploid and subgenomes of tetraploid species (Supplementary Fig. 45). However, this analysis of the entire set of ohnologs may obscure more subtle differences. To examine this, we next analyzed CG methylation for genes with biased expression in muscle tissue towards either the paternal subP or maternal subM. Interestingly, CG methylation levels of expression biased genes towards the subgenome A were lower from ~1.5 kb upstream to TSS compared with subM levels (Supplementary Fig. 46a). Similarly, the upstream region of subM bias genes for all species showed lower CG methylation levels in this same region than those of the corresponding regions of duplicated genes in subP (Supplementary Fig. 46b). This suggests that CG methylation levels in upstream regions of genes may have a role in observed expression bias towards a particular subgenome.

Further, to determine if there are any significant differences in TE CG methylation between subP and subM of tetraploid species, we investigated CG methylation of TEs that are in 1 kb vicinity of 7040 positionally conserved syntenic ohnologs and at the whole genome level. We found some degree of variation in mCG levels between subgenome TEs that were found in 1 kb vicinity of 7040 duplicate orthologs (Supplementary Fig. 47a). However, elevated levels of TE methylation in subP were observed in *L. capito* which was opposite to what was observed in *S. sinensis* and *P. rabaudi* where subM showed higher methylation levels. This phenomenon was also observed for TE methylation at the whole genome level (Supplementary Fig. 47b). This opposite trend of TE methylation in *L. capito* in comparison to *S. sinensis* and *P. rabaudi* can be attributed to the difference in the TE density of respective genomes (Supplementary Fig. 7).

### Subgenome dominance patterns in three allotetraploids - Genome Architecture

Finally, we investigated the three-dimensional (3D) genome architecture of each allopolyploid, including subgenome compartments and topological associated domains (TADs) using Hi-C data (Supplementary Tables 6–8). Our results indicate that the genome occupation of open-chromatin regions ("A compartments") ranged from 46.5% to 48.6% in each subgenome and was less than those of closed-chromatin regions ("B compartments") which ranged from 51.4% to 54.0% (Fig. 6a, Supplementary Figs. 48–50, Data 5). A similar pattern was also observed for the TE content in the A and B compartments (Fig. 6b). Roughly 61% genes of each subgenome were found to be associated with A compartments, and the remaining 39% of genes with B compartments (Fig. 6c, Supplementary Data 5). As expected, we found that the B compartment genes showed significantly lower expression levels when compared with those in the A compartments (a two-sample t-test; $p < 0.0001$) (Fig. 6d). We also demonstrated that the ratio of nucleotides, TEs and genes in A/B compartments in subP of each allotetraploid displayed no clear difference with those in subM (Fig. 6a–c, Supplementary Data 5).

Vertebrate genomes including zebrafish are organized into Topological Associated Domains (TADs)[70]. Here, hicFINDTAD[71] and HiTAD[72] were used to explore and annotate TADs across the 3D genome of the three allotetraploids. The hicFINDTAD[67] method identified 2782, 3237 and 3383 TADs in *S. sinensis*, *P. rabaudi* and *L. capito*, respectively, whereas fewer TADs were obtained using HiTAD[72] in *P. rabaudi* (2620) and *L. capito* (2920) (Fig. 6e, Supplementary Table 22). The identified TADs by these two methods were compared between subgenomes of each allotetraploid (Fig. 6f, Supplementary Table 22). The size of TADs ranged from 80 kb to 5 Mb, averaging about 470 kb in the four subgenomes of *P. rabaudi* and *L. capito*, and about 25% higher (average 586 kb) in subP and subM of *S. sinensis* (Supplementary Table 23). The gene number in TAD boundaries and TAD sizes were similar between each subgenome (a two-sample t-test; no significant) (Fig. 6g and h). Together, this study suggests that the 3D chromatin structure is conserved between the two subgenome of each allotetraploid.

## Discussion

Cypriniformes represent the largest clade of freshwater fish with ~600 described species in the family Cyprinidae, which has experienced multiple rounds of independent WGD[34]. The phylogenetic relationships, evolutionary history, and the genetic basis of previously reported subgenome dominance of these polyploids has remained poorly understood. In this study, high-quality genomes of twenty-one cyprinid fishes, including subgenome-resolved allotetraploid genomes from three tribes, were de novo assembled and analyzed to investigate subgenome evolution at the genetic and epigenetic levels. Our results are supportive of previous reports for subgenome dominance at both the gene retention and transcriptome level[29]. In addition, we observed that the dominant subgenome retained a greater number of tandem duplicates with a functional bias towards immune related processes.

Our phylogenetic analyses revealed that *S. sinensis*, *L. capito*, and *P. rabaudi* are allopolyploids and that observed dominance is consistently towards the subgenome contributed by the maternal parent. Also, the most recent polyploid event in *P. rabaudi* is likely shared with common carp and goldfish. Functional enrichment analyses revealed similar GO term classes, including mitochondrial related processes, for the genes that returned to single copy in all examined allopolyploids. The observed consistent bias towards the maternal subgenome donor, alongside the bias towards mitochondrial functions, suggests that observed subgenome dominance patterns in these allopolyploid fish may be due to maternal dominance. The maternal contributed nuclear-encoded genes that interact with mitochondrial encoded genes may be favored to maintain proper cytonuclear interactions[73].

The mitochondrial proteome contains products from over a thousand genes, while the mitochondrial genome encodes approximately only 13 proteins (i.e. 1% of the proteome)[74]. The vast majority of genes are now nuclear genome encoded following the horizontal gene transfer from the organellar genome to the nuclear genome over the past hundred million years[75]. However, these nuclear genes might encode dosage-sensitive proteins that function in either organellar signaling networks or macromolecular complexes that must maintain proper stoichiometric balance with interacting partner (s) that are encoded in the organellar genome[76]. Furthermore, the sequence of the proteins encoded by both organellar and nuclear-encoded mitochondrial genes may have diverged among the diploid progenitors. Thus, there's a possibility for incompatibilities to arise from "mismatches" between the genes contributed by the paternal subgenome and the organellar genomes contributed by the maternal parents in allopolyploids[77]. The biased expression of the maternal nuclear copy would resolve any potential conflicts. The model that we are proposing here is that observed dominance patterns in these allopolyploids is to preserve proper cytonuclear interactions, and ultimately, core cellular functions. Nonetheless, we cannot exclude the possibility that some of

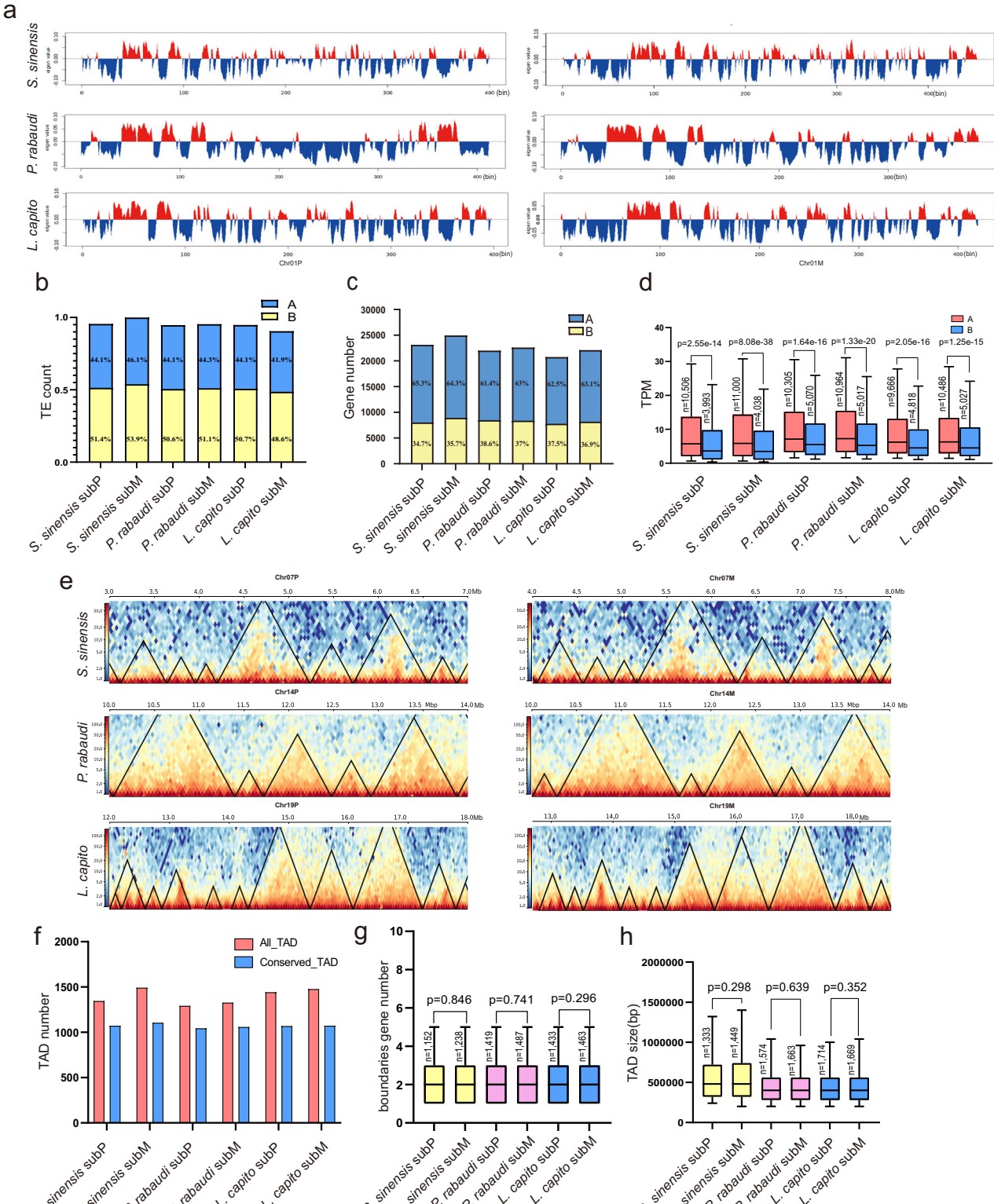

**Fig. 6 | Three-dimensional (3D) genome architectures, including A/B compartments and topological associated domains (TADs), of each subgenome from three allotetraploids. a** First principal component values representing A/B compartments in the Chr01P and Chr01M of *S. sinensis*, *P. rabaudi*, and *L. capito*. Positive PC values showing in red are designated as A compartments, and negative PC values indicating in blue represent B compartments. A/B compartments found in the rest chromosomes of three species are shown in Supplementary Figs. 45–47. **b** TE content in A/B compartments in subP and subM. **c** Gene number in A/B compartments in subP and subM. **d** Expression level of genes in A compartments was significantly higher than those in B compartments (Two-sample t-test;

*p* < 0.001). **e** TAD structure in one representative region of homologous chromosomes from three allotetraploids. Black triangles show TADs. Yellow blocks indicate strong signal of chromatin interactions and blue blocks indicate weak signal of chromatin interactions. **f** Number of TAD (red) and conserved TAD (blue) identified in each subgenome. **g** Gene number in TAD boundaries in each subgenome. **h** TAD size in each subgenome. All the t statistical test used in this figure was two-sided, and the exact p values were also showed. It indicates non-significant in two-sample t-test results if p values were larger than 0.05. The sample size used for statistical analysis is shown as "n".

the observed subgenome expression differences, particularly at the individual gene level, is due to differences in DNA methylation and transposable element density differences as hypothesized in previous studies[19].

We observed that methylation levels at CG sites in upstream regions of genes, -1.5 kb upstream to the transcriptional start site, may have a role in observed expression bias towards the maternal subM genome. Epigenetic factors, including changes in methylation at certain CG sites, have been previously shown to alter gene expression and involved in maternal imprinting including of nuclear encoded mitochondrial and DNA repair genes[78,79]. We also observed that the dominantly expressed ohnolog, from either subgenome, in some cases, had significantly lower TE densities. This suggests that both maternal dominance and TE differences are likely contributing to observed independently repeated subgenome dominance patterns in allopolyploid cyprinid fishes. To the best of our knowledge, this is the first study to show the potential role of maternal dominance in contributing to subgenome dominance in any allopolyploid animal. Future studies of other allopolyploids are needed to determine if these observed patterns are shared by other polyploid animals or are potentially unique to cyprinids.

Furthermore, our multi-species comparisons suggest that genetic divergence of the diploid progenitors, for the allopolyploids and divergence times examined in this study, did not contribute to subgenome expression dominance. However, it is important to note the possibility that the divergence of the diploid species in each allopolypoid wasn't sufficiently different to observe additive subgenome expression dominance effects. Lastly, we also examined genome organization using Hi-C data and selective constraints on noncoding regulatory sequences, which revealed no significant differences among subgenomes. These new reference genomes and various datasets should serve as a powerful platform for the community to further investigate genome evolution of cyprinids, and as a valuable resource for a wide range of studies including modeling human disease[80].

# Methods

## Samples

The collection and use of fishes here were done according to the permissions for scientific purposes set up by the Animal Care and Use Committee of Jiujiang University (ID:JJU20200042). No permits were required for the collection of wild fish. The detail information of 21 cyprinid fishes used in this study is listed in Supplementary Table 1. All fishes were anesthetized with clove oil. Muscle genomic DNA of these fishes was chosen for DNA sequencing. RNA-Seq was also performed on 2 - 12 tissues for each cyprinid fish (Supplementary Data 6) and the different tissues of each species were pooled for Iso-Seq. To test subgenome dominance and functionalization mechanisms of retained duplicated ohnologs, 3-5 biology replicates of six tissues (brain, eye, gill, heart, liver and muscle) from three allotetraploids *S. sinensis*, *L. capito*, and *P. rabaudi* and two possible diploid ancestors *O. macrolepis*, and *Sc. acanthopterus* were collected for RNA-Seq. Muscle with five biology replicates of these five species was also used for whole-genome bisulfite sequencing. All tissue samples were first frozen in liquid nitrogen and then stored at −80 °C until analysis.

## Genome sequencing

For genome survey, 21 genomic DNAs were extracted from muscle tissues with the Qiagen DNeasy kit. Paired-end libraries were constructed with insert sizes around 350 bp according to Illumina's protocol and were sequenced using the Illumina NovaSeq platform. Approximately 50.8–153.7 Gb of raw Illumina short-read data were generated. The raw reads were first cleaned by filtering out the adapter sequences, low-quality bases and reads. After this process, 50.5–153.2 Gb of clean bases were retained (Supplementary Table 2).

Then the short-reads data was used to calculate K-mer frequencies (K = 17) and distribution with Jellyfish v.2.2.6[81] which the genome size was estimated to be 0.83–1.84 Gb with heterozygosity of 0.26–0.99% and a repetitive DNA content of 44.55–66.05% (Supplementary Table 4).

HiFi sequencing of these 21 fishes began with DNA samples that were randomly broken into fragments by g-TUBE (Covaris) and large fragments (≥ 20 kb) of DNA were enriched and purified by magnetic beads. Fragments of DNA were repaired by damage repair and end repair. The stem ring-like adaptors were added at the two ends of DNA fragments, and the fragments that failed to be repaired were removed by exonuclease. SMRTbell long read libraries (-40 kb) were prepared according to the released protocol from PacBio. The single-molecule real-time (SMRT) sequencing was performed on the Pacific Biosciences Sequel II platform, in the circular consensus sequencing (CCS) mode. A total of 20.15–59.91 Gb of HiFi reads was generated using CCS (https://github.com/PacificBiosciences/ccs) for each of the 21 newly sequenced fishes (Supplementary Table 2).

## HiC sequencing

For the construction of HiC libraries, muscle DNA of *S. sinensis*, *L. capito*, and *P. rabaudi* was extracted from in vitro seedlings, which were digested with HindIII using the previously described HiC library preparation protocol[82]. These HiC libraries were sequenced on an Illumina NovaSeq platform.

## RNA-Seq

The total RNA from each tissue was extracted for the library construction. These libraries were subsequently sequenced on the Illumina HiSeq platform, which produced around 6 Gb data for each tissue in each sample. For full-length transcriptome sequencing, a mixed RNA library from different tissues was prepared according to the PacBio ISO-Seq experimental workflow and subsequently run on a PacBio Sequel II platform.

## MethylC-Seq and analysis

Genomic DNA degradation and contamination was monitored on agarose gels. DNA purity was checked using the NanoPhotometer spectrophotometer (IMPLEN, CA, USA). DNA concentration was measured using Qubit DNA Assay Kit in Qubit 2.0 Flurometer (Life Technologies, CA, USA). Microgram genomic DNA spiked with lambda DNA were fragmented by sonication to 200-300 bp with Covaris S220, followed by end repair and adenylation. Cytosine-methylated barcodes were ligated to sonicated DNA as per manufacturer's instructions. Then these DNA fragments were treated twice with bisulfite using EZ DNA Methylation-GoldTM Kit (Zymo Research), before the resulting single-strand DNA fragments were PCR amplified using KAPA HiFi HotStart Uracil + ReadyMix (2X). Library concentration was quantified by Qubit 2.0 Fluorometer (Life Technologies, CA, USA) and sequenced by Novaseq platform w producing 24.39-55.95 Gb raw bases with a bisulfite conversion rate of 99.57–99.75%.

MethylC-seq data for each sample were aligned to their respective genomes and methylation called using the methylpy pipeline v.1.4.6[83]. This pipeline uses Cutadapt v.4.1[84] for adapter trimming, Bowtie 2 v.2.4.4[85] for alignment, and Picard v.2.26.10 (https://broadinstitute.github.io/picard/) to mark duplicate reads. Spiked-in unmethylated lambda phage DNA was used as a control to calculate non-conversion rates from bisulfite treatment (Supplementary Table 21). Gene and TE metaplots were made as previously done[86] using custom scripts (https://github.com/niederhuth/methylation_scripts) and pybedtools v.0.9.0[87]. Gene/TE bodies were divided into 20 bins, and the weighted methylation level[88] calculated across all genes/TEs. For gene bodies, only exonic cytosines were included. This process was repeated for both 2 kb upstream and downstream regions, and the data plotted in R with ggplot2. To examine the effects of neighboring TEs on genic

methylation, we used bedtools v.2.30.0[81] to identifying genes with an intersecting TE within 1 kb.

## Genome assembly, purging and evaluation

The initial assemblies of 21 genomes were performed using Hifiasm v.0.16.0[42] with "-k 51 -D 5.0 -r 3 -l 3" settings on the 20.15–59.38 Gb raw PacBio HiFi reads and purged by purge_haplotigs (https://github.com/skingan/purge_haplotigs_multiBAM). Approximately106 Gb, 117 Gb, and 102 Gb of raw Hi-C data were obtained in *L. capito*, *P. rabaudi*, and *S. sinensis*, respectively. To build pseudo-chromosome level genome, the cleaned Hi-C reads were first mapped to the primary contig by BWA v.0.7.17[89] mem with "-k 32 -w 10 -B 3 -O 11 -E 4" parameters. Next, the draft genome was scaffold with Hi-C reads by the ALLHiC algorithm[41]. Then, the Juicebox tool[90] was used to manually adjust chromosome construction and assembly errors. Finally, we anchored the contig of three species on fifty chromosomes respectively. The BUSCO v3.0.2 pipeline[43] was used to assess the genome completeness with the actinopterygii_odb9 dataset, which contains 4584 gene sets. Short reads generated by the Illumina NovaSeq platform of each accession were aligned to the genomes using BWA v.0.7.10-r789[91]. The unique mapping data were used for identifying single nucleotide polymorphisms (SNPs) and InDels using SAMtools v.0.1.19[92] to assess sequencing errors in the long-read data as well as estimate within-individual rates of heterozygosity.

## Subgenome identification

We partitioned the *L. capito*, *P. rabaudi*, and *S. sinensis* genome into subP and subM by SubPhaser pipeline[52]. First, we counted the number of 15-mers sequences using Jellyfish v.2.2.6[81] for each chromosome. Next, the different kmers were identified among homoeologous chromosome groups, and cluster into subgenomes by a K-Means algorithm. Finally, the genome was phased to subP and subM by hierarchical clustering and principal component analysis (Supplementary Fig. 10).

To identify subgenome-specific TEs, we implemented subgenome biased index (SBI) analysis[39]. TEs were identified using RepeatMasker v.4.1.0 (https://www.repeatmasker.org/). Their numbers on the $i^{th}$ chromosome of the subP and subM were counted as $N_{iA}$ and $N_{iB}$ respectively. For each given TE, its SBI was then calculated as: SBI ranges from 0 to 1 with values close to 1 indicating the TE exhibits subgenome biased distribution. The homoeologous chromosomes were determined based on gene conserved synteny between each allotetraploid and *O. macrolepis* using the MCscan pipeline[93].

## Repeat annotation and TE age analysis

TEs were identified using a combination of de novo and structure-based methods. First, RepeatModeler v.2.0.2a[94] was used for de novo annotation of the transposons. Second, LTRharvest v.1.5.10[95] and LTR_Finder v.1.07[96] were used to identify the LTRs by structure prediction, and LTR_retriever v.2.9.0[97] was used to integrate the prediction results gained by LTRharvest[95] and LTR_Finder[96] to obtain high-quality LTR sequences. Third, EAHelitron v.1.5.3[98] was used to identified Helitron and MITE-Hunter v.1[99] was used to found miniature inverted repeat transposable elements (MITEs). Then, we combined transposons obtained from the above multi-tool joint prediction method into a library. For each species, extracted the transposons which are classified to superfamilies and divided them into five categories: DNA, LTR, LINE, SINE, and Unknown. We use cd-hit (-c 0.8) v.4.8.1[100] to de-redundant these data respectively, and the obtained results were reclassified using REPCLASS v. 1.0.1[101]. Transposons still in the unknown state were classified again using TECLSS[102]. Finally, we used our obtained TE database as queries to BLASTx against the SwissProt database to remove non-TE host genes (E-value ≤ 1e-10). The final TE library for each species was used to mask the assembly genome using RepeatMasker (http://www.repeatmasker.org/, v.4.1.2) to calculate copy number and content.

The ages of individual TE insertions were inferred based on the methods in Chang et al. 2022[50] which, in short, builds a tree with insertions from a TE family and uses the terminal branch lengths as a proxy for the age of the insertions. First, we trimmed the repeat annotations (.gff files) from RepeatMasker removing any insertions <50% of the length of the full length TE. We then used bedtools v.2.26.0[103] getfasta to extract the sequence of individual TE insertions with the -s flag for strand awareness. To ensure that subsequent steps (alignment and tree-building) complete in reasonable time, for families with over 500 insertions, we down-sampled to only 500 insertions, and removed families with fewer than 5 insertions. As per Chang et al. 2022[50], for each family, we used MAFFT v7.490[104] to align the sequences of TE insertions generating multiple sequence alignments which were then inputted into TrimAl v1.4.1[105] with the parameter -gt 0.01 to trim the alignments. We used FastTree v2.1.11[106] with the parameters -nopr -nt -gtr to construct the phylogeny. The trees were then loaded into R and processed with the phytools package[107] in R to determine the terminal branch length of each insertion in the tree. The age of a family/superfamily/class is the median taken across all insertions.

Simple Sequence Repeats (SSR) analyses were determined using GMATA v.21 (https://sourceforge.net/projects/gmata/) with parameter -r 5 -m 2 -x 10 -s 0. Tandem Repeat Finder (TRF) v.4.09[108] with parameter 2 7 7 80 10 50 2000 -d -h 1 were used to predict tandem repeats.

## Non-coding gene annotation

The tRNAscan-SE v.2.0[109] was used for de novo annotation of tRNAs. The rRNA and other non-coding RNAs were annotated by Rfam v.14.1[110].

## Protein coding gene prediction and annotation

After masking TEs, SSRs and tandem repeats, three approaches for prediction of the protein-coding genes of our assembly genomes were employed, including homology-based prediction, ab initio prediction, and transcriptome-based prediction methods. For protein-homology-based prediction, the protein sequences of five species, including *Ancherythroculter nigrocauda*, goldfish, zebrafish, *Onychostoma macrolepis* and common carp were used. The five species were aligned against the assembly genomes using tBLASTn v.2.9.0 (E-value = 10e-5)[111]. GeMoMa v.1.6.1[112] was used to predict the exact gene structure of the corresponding genomic region on each BLAST hit. For transcript-based prediction, RNA-seq were used to align by Hisat2v.2.2.1[113] and assembly into transcript with Cufflinks v.2.2.1[114]. The transcript and ISO-seq were merged and de-redundancy by StringTie v.2.1.6[115] with the –merge parameter. These assembled sequences were then aligned against the genome using Program to Assemble Spliced Alignment (PASA) v.2.5.2[116] for supporting the reference genome annotation. We simultaneously employed four tools of Augustus v.3.2.2[117], SNAP v1[118], GlimmerHMM v.3.0.4c[119], and GeneMark-ETv1[120] for ab initio prediction, which was trained by PASA-H-set gene models. According to these three approaches, all the gene models were finally integrated by EvidenceModeler v1.1.1[121]. The BUSCO v.3.0.2 pipeline[43] was used to evaluate the gene completeness with the actinopterygii_odb9 dataset, which contains 4584 gene sets.

## Mitochondrial genome assembly and phylogenetic tree

The mitochondrial genome of each species was assembled by NOVO-Plasty v.4.3.1[122], with the following procedure: (1) the mitochondrial genome of zebrafish was downloaded from GenBank as a starting reference; (2) The cleaned next-generation sequencing data from each species was used by "NOVOPlasty.pl", with the parameters "Type = mito, Genome Range = 12000-21000, K-mer = 31, Read Length = 151, Insert size = 350, Single/Paired = PE, Insert size auto = yes, Insert Range = 1.9, Insert Range strict = 1.3". The assembled mitochondrial

genomes were annotated by MEANGS v.1.0[123] with the parameter "−skipassem". As no mitochondrial genome was available for *Danionella translucida*, we obtained its Illumina sequences (SRR8713016) to reconstruct its mitochondrial genome using the above same process. Mitochondrial genomes were aligned by MUSCLE v.5.1[124] and trimmed by trimAl v.1.4[105] using the option "nogaps". Our 22 assembled and 15 reported mitochondrial genomes (Supplementary Table 24) were used to build ML phylogeny using RAxML v.8.2.12[125], with mean bootstrap percentages (BPs) computed with 1000 replicates using the "GTR + GAMMAIX" model for the rapid bootstrapping algorithm (Supplementary Fig. 14).

## Ortholog identification

Orthologous genes were identified by sequence similarity clustering followed by phylogeny analysis. In detail, we first pooled together the protein sequences of all genes and ran an all-vs.-all blast using blastp v.2.2.28+[111] with an e-value cut-off at $1 \times 10^{-3}$. So for each two genes, an H-score[126] was calculated from the blast raw score to index the extent of sequence similarity. With the H-score as the "distance", all genes are clustered and divided into different groups (gene families) using Hcluster_sg v.0.5.1[127]. In each group, protein sequences were aligned using MAFFT v.7.453[104] with default parameters and transformed into coding sequence alignment using PAL2NAL v.14[128]. A gene tree was then reconstructed based on the CDS alignment using TreeBeST v.1.9.2 (https://github.com/Ensembl/treebest) with default parameters. Accordingly, the orthologous relations between genes were determined as n to m (n and m are positive integers with cases where n = m).

## Phylogenomic analysis

A phylogenomic tree of species and subgenome relationships was reconstructed based on single-copy orthologs across species/subgenomes. In total, 300 gene orthologs with a 1:2 relatiosnhip (one copy in diploids, and one copy on the subP and one copy on the subM) were obtained from our sequenced 21 species and 16 reported species using the above method. Then, we aligned the protein sequences of the single copy orthologs using MAFFT v.7.453[104] and trimmed the alignment using trimAl v1.4[105] with default setting. The alignments were concatenated into a large alignment, and was then transformed into CDS alignments using PAL2NAL v.14[128].The result was transferred into IQ-TREE v.2.0.3[129] with the parameters of –boot-trees -B 1000 -m MFP for constructing the maximum likelihood (ML) phylogenies (Supplementary Figs. 13). To estimate the effect of different outgroups to the topology structure, a ML tree (Supplementary Figs. 14) was rebuilt using RAxML v.8.2.12[125] based on CDS from 310 gene orthologs from 36 species with a 1:2 relatiosnhip, which were determined using the ortholog identification method mentioned above and implemented a strict synteny confirmation where at least five orthologs have to be arranged in a row with the largest gap being fewer than five genes.

We also obtained gene orthologs with a 1:2 relatiosnhip of 13 species using this similar method to further determine the relationship between diploid ancestors and subgenomes from five allotetraploids (*L. capito*, *P. rabaudi*, *S. sinensis*, goldfish, and common carp). 1669 gene orthologs were found from these species and their protein alignments were concatenated, and was then transformed into CDS alignments using PAL2NAL v.14[128]. A ML tree based on CDS was built using IQ-TREE v.2.0.3[129] with the parameters of –boot-trees -B 1000 -m MFP (Fig. 3b). Meanwhile, Four-Fold Degenerate Sites (4DTV) were extracted from CDS alignments using an in-house perl script, after which 252,437 4DTVsites were extracted. RAxML v.8.2.12[125] was then run to construct the ML phylogeny using these 4DTVsites. The bootstrap values of nodes were all achieved 100% (Supplementary Fig. 17). To construct the whole-genome alignment (WGA) of these 13 species, we first built the pairwise genome alignments of each species to *O.*

*macrolepis* using minimap2[130] with parameter "-cx asm20 −cs=long". The alignments were then improved using Genome Alignment Tools from Hiller lab (https://github.com/hillerlab/GenomeAlignmentTools). In detail: axtChain was used first to chain up the alignment blocks, the unaligned flanked loci were then re-aligned using patchChain.perl. RepeatFiller was then used to detect and incorporate the repeat-overlapping alignments. Obscure local alignments were removed using chainCleaner. Finally, chainNet was used to collect alignment chains hierarchically to capture only the orthologous alignments. Those improved pairwise genome alignments were merged into the multiple WGA using MULTIZ (https://www.bx.psu.edu/miller_lab/). The final WGA achieved ~120 Mb. To reconstruct the phylogenetic tree from the WGA, we only kept those alignment blocks longer than 1 kb. The rest alignments were trimmed using trimAl v1.4[105] and concatenated into an alignment around 26 Mb. RAxML v.8.2.12[125] was used for a maximum likelihood inference of the phylogenomic tree under GTR+Gamma model with 1000 rapid bootstraps. The bootstrap values of nodes were all achieved 100% (Supplementary Fig. 16).

In addition, multiple sequence alignments of 1669 genes of 13 species were constructed using MUSCLE v.3.8.31[124]. The best fitting tree per gene was obtained using RAxML-NG v.1.1.0[131] from 20 inferred trees under GTR + GAMMA with default parameters. Bootstrap values of 50 replicates were mapped onto the best-scoring ML-tree. Genes for which bootstrapping did not converge were filtered out, retaining 1665 genes. Consensus trees across these gene trees were visualised using DensiTree[132]. Finally the Bayesian posterior probability support (based on the amount of gene trees supporting the split) for each clade was calculated and plotted on top of the consensus tree using SumTrees, part of DendroPy[133] and visualised using FigTree v.1.4.4 (http://tree.bio.ed.ac.uk/software/figtree/) (Supplementary Fig. 18).

## Divergence time calibration

MCMCTree v.4.9e[134] under a relaxed-clock model (correlated molecular clock) was used to estimate the time of divergences for the phylogenetic tree of 300 gene orthologs with a 1:2 relatiosnhip built by IQ-TREE v.2.0.3[129]. First, based on the phylogenomic tree and CDS alignment the substitution rate was roughly estimated using baseml. Then, we run mcmctree for the first time to estimate the gradient and Hessian. The result was output into the file out.BV and used by the final run of MCMCtree v.4.9e[134] to perform approximate likelihood calculations (Supplementary Fig. 12). During the process, the Markov chain Monte Carlo process was run for 2,005,000 steps. We discarded the first 5,000 steps as burn-in and did the sampling every 100 steps to have 20,000 samples collected. Four time calibrations were set: 81.9-100.1 MYa to the most recent common ancestor of Nemacheilidae and Cyprinidae[135]; 27.82-33.9 MYa to the most recent common ancestor of Barbinae and Cyprininae[136]; 40.4–48.6 MYa to the root of Cyprinidae[136–138]; and 11.6–12.7 MYa to crown age of Leuciscinae[139]. This phylogenetic tree was visualized in ITOL (https://itol.embl.de/) (Fig. 1).

## Dating of divergence

Pairwise *Ks* values of orthology gene pairs were used as molecular clocks to date the divergence of species/subgenomes. After further confirmation of synteny conservation that requires at least five genes arranged in a row with the largest gap being fewer than five genes, we identified 13264, 15935, 14104, 16349, 16219 and 16205 pairs of orthologous/ohnologous genes for subP and subM of *L. capito*; subM of *L. capito* and *S. acanthopterus*; subP and subM of *P. rabaudi*; subM of *P. rabaudi* and *Sc. acanthopterus*; subP and subM of *S. sinensis*; and subM of *S. sinensis* and *O. macrolepis* respectively. For each pair of the orthologs/ohnologs, their protein sequences were aligned using MAFFT v.7.453[104] and then transformed into coding sequences (CDS) using PAL2NAL v.14[128]. Alignment gaps were removed using Gblocks

0.91b[140]. The pairwise *Ks* values were then calculated based on the CDS alignments using codeml in PAML v.4.9e[134]. Median values of *Ks* were used to calibrate the divergence events (Fig. 3a).

### Identifying core, softcore, dispensable, and private genes
Based on the results of gene family clustering (please see the part **Ortholog identification** for detail), the core, softcore, dispensable, and private genes of cyprinid fishes were classified by using a Perl script. The gene families shared by all 36 species (*T. bleekeri* was not considered since it do not belong to cyprinid fishes) were defined as core genes, the families existed in 32-35 species were defined as softcore genes, the families distributed in 2-31 species were defined as dispensable genes, and the families only presented in one species were defined as private genes (Supplementary Fig. 37). Besides, allopolyploidy, a gene family appears in either subP or subM, we considered that the allopolyploid species contain this gene.

### Homebox gene identification and classification
We isolated putative Homeobox genes in three allotetraploids by performing tBLASTn v.2.9.0 (E-value = 10e-5)[111] using *Homo sapiens*, *Mus musculus* and zebrafish Homeobox protein sequences curated from NCBI as queries. The BLAST hits were then conjoined by Solar software[141]. GeneWise v.2.4.1[142] was used to predict the exact gene structure of the corresponding genomic region on each BLAST hit. HMM v.3.2 searches of the above genes against the Pfam database (http://pfam.xfam.org/) revealed having Homeodomain to each gene. The classification of deduced proteins and their integrity were verified by building genes tree with zebrafish Homeobox protein sequences, through MAFFT v.7.490[104] and FastTree v.2.1[106].

### Expression comparison between ancestors and allotetraploid ohnologs
To investigate evolutionary mechanism for duplicated genes of three allotetraploids, 7040 genes with a 1:1:2:2:2 relationship (1 *O. macrolepis* gene, 1 *Sc. acanthopterus* gene, 2 *S. sinensis* genes, 2 *L. capito* genes and 2 *P. rabaudi* genes) were obtained using our ortholog identification method. RNA-seq reads sequenced by illumina NovaSeq platform from six shared tissues (muscle, gill, heart, eye, brain, liver) with three to five replicates were mapped to respective genome assembly using Hisat2 v.2.2.1[113]. Expression levels (TPM) were estimated using String-Tie v.2.1.6[115]. A gene was said to be expressed if TPM ≥ 1 in at least one tissue. orthologs or ohnologs without expressed genes in all fishes were removed. The Pearson's correlation coefficient of expression patterns between ancestor and individual allotetraploid ohnologs and between ancestor and ohno-pair was used to detect expression correlation. Two genes were denoted as highly correlated if the Pearson's correlation coefficient between their log2(TPM + 1) was greater than 0.75 and with correlation test (cor.test in R) P < 0.1, medially correlated if their correlation coefficient was greater than 0.6, and differentially expressed if the t test between their log2(TPM + 1) was less than 0.01. We defined that gene A is "on" relative to gene B if TPM(A) ≥ 2 and TPM(B) < 1 and identified coexpressed, nonfunctionalized, subfunctionalized, and neofunctionalized orthologs following on-off conditions as described[37]. The results were listed in Supplementary Table 15 and Fig. 20.

### Gene fractionation
The genomes of three allotetraploid Cyprinidae species (*L. capito*, *P. rabaudi*, and *S. sinensis*) were each aligned to three diploid references (zebrafish, *O. macrolepis*, and *Sc. acanthopterus*) with the LAST alignment algorithm[143] in CoGe'sSynMap[144]. Neighboring syntenic blocks with a maximum distance of 40 genes were merged with the Quota Align Merge option[145]. A maximum of 50 query chromosomes was set for each allotetraploid, whereas a maximum of 25 reference chromosomes was set for each diploid reference, with the ratio of coverage

depth set to 1:2 (Fig. 4a, Supplementary Figs. 21–29, Table 16). All fractionation bias analyses can be regenerated at the CoGe links provided under URLS.

*L. capito* v. zebrafish: https://genomevolution.org/r/1m3g7
*L. capito* v. *O. macrolepis*: https://genomevolution.org/r/1m3gp
*L. capito* v. *Sc. acanthopterus*: https://genomevolution.org/r/1m3go
*P. rabaudi* v. zebrafish: https://genomevolution.org/r/1m3d0
*P. rabaudi* v. *O. macrolepis*: https://genomevolution.org/r/1m570
*P. rabaudi* v. *Sc. acanthopterus*: https://genomevolution.org/r/1m571
*S. sinensis* v. zebrafish: https://genomevolution.org/r/1m2og
*S. sinensis* v. *O.macrolepis*: https://genomevolution.org/r/1m24h
*S. sinensis* v. *Sc. acanthopterus*: https://genomevolution.org/r/1m2b8

### Tandem duplicate identification and analysis
Tandem duplicated genes were identified using SynMAP[144] as described above with the tandem duplication setting set to 10. Results can be regenerated using the above weblinks. The aforementioned ortholog predictions to Zebrafish genes for identified tandem duplicated genes in each allopolyploid (*L. capito*, *P. rabaudi*, and *S. sinensis*) were analyzed in the STRING v.11.5[146] to perform PFAM enrichment analysis. The p-value was computed using a hypergeometric test in STRING v.9.1[147], and multiple testing was corrected using the method of Benjamini and Hochberg[148]. Tandem duplicate summary tables and PFAM enrichment analysis results are available in Supplementary Table 17; Fig. 5.

### Gene expression fractionation
Global homoeolog expression bias (HEB) was analyzed as the average TPM of three to five replicates for all genes of a species for a given tissue type (Supplementary Fig. 35). Subgenome assignment was based on chromosome names reflecting subgenome identity. Homoeolog expression bias of syntelogs was plotted as the number of syntenic orthologs with biased expression (Fig. 4b, Supplementary Fig. 39), defined as TPM log$_2$fold change > |2| following methods in Woodhouse et al. 2014[25]. Significant departures from balanced gene count were determined via chi-square statistics (Supplementary Tables 18 and 19).

### TE density analysis
In order to investigate subgenome TE biases, TE Density v.1.0.1[149] was run independently with default options for *L. capito*, *P. rabaudi*, and *S. sinensis*. TE Density calculates the proportion of TE-occupied base-pairs relative to genes and a given window measurement size. TE density was calculated for the combination of (TE superfamily identity ‖ TE order identity) × (upstream ‖ intragenic ‖ downstream), with a window length of 10,000 bp up and downstream of gene start and stop positions. Gene and TE annotation files, the inputs to the software, were reformatted to conform to the requirements of TE Density. Analyses were conducted and graphs were generated using Python v.3.8.0. Version-controlled code (Fig. 4c, Supplementary Fig. 40) and documentation related to this analysis can be found at https://github.com/sjteresi/Fish_TE_Differences, see the requirements directory in the project GitHub repository for a more complete list of minor packages. Genes which displayed a bias in expression to either subgenome in at least three tissue-types were further investigated using TE Density. A two-sample t-test was calculated for subP and subM TE density values in the sets of biased genes for each subgenome and focal allotetraploid (Supplementary Figs. 41 and 42).

### Constraint on CNSs
A set of CNSs was developed using the Haudry et al. 2013[150] pipeline with minor modifications for a vertebrate setting, briefly: a set of 11 diploid species (*Sinilabeo rendahli*, *Crossocheilus oblongus*,

zebrafish, *Gobiobotia tungi*, *Rhodeus sinensis*, *Zacco platypus*, *Mylopharyngodon piceus*, *Sc. acanthopterus*, *P. huangchuchieni*, *Puntius tetrazona* and *Semilabeo prochilus*) were soft repeat-masked using their associated gff repeat annotations where available, and where not, were repeat modeled and soft masked ([http://]www.repeatmasker.org, repeatmasker v.4.0.6, repeat-modeler v.2.0). Only these diploid species were used to infer CNSs due to the potential for constraint loss in polyploids. Soft-masked genomes were pairwise-aligned to the *O. macrolepis* reference genome using lastz (https://github.com/lastz/lastz, v.1.04.03) with settings –gapped –nochain –strand=both –step=10 –ambiguous=iupac –format=axt as were 5 tetraploid species (goldfish, *P. rabaudi*, *S. sinensis*, common carp and *L. capito*). Alignments were chained using axtChain (Kent utils, https://github.com/ENCODE-DCC/kentUtils) in accordance with vertebrate recommendations (http://genome.ucsc.edu/cgi-bin/hgTrackUi?db=danRer7&g=vertebrateChainNet) lineargap=loose and minScore=5000. Chains were sorted by score and candidate best orthologous chains were selected from the highest scoring chains for each region of the reference and non-reference genomes (see Haudry et al. 2013[150]).

For each species chains were converted to axt format (Kent utils, chainToAxt) and from axt to pairwise maf format (Kent utils, axtToMaf). Maf files were sorted and combined in order of approximate phylogenetic distance from the reference (closest first) using multiz (https://genome.cshlp.org/content/14/4/708.full, v.11.2) to generate a 12-way maf multiple alignment. A neutral phylogenetic tree was generated in a two-step process, initially the MAF blocks with all 12 species present were extracted to a separate file and RAxML v.8.2.12[125] in GTRCAT mode was used to generate the phylogenetic relationships, then 4D sites were extracted from the MAF alignment using msa_view (https://github.com/CshlSiepelLab/phast, v.1.3) and the *O. macrolepis* gene annotation, and phyloFit (https://github.com/CshlSiepelLab/phast, v1.3) was used to refine the neutral branch lengths in the RAxML tree using only the 4D sites[150]. With this model as a guide to the neutral state, phastCons (https://github.com/CshlSiepelLab/phast, v.1.3) was used to infer non-neutral conserved regions using the options –msa-format MAF –target-coverage 0.05 –expected-length 20 –rho 0.2[151]. A threshold phastCons score of 0.82 was applied to determine conserved regions of the genome. In order to remove from consideration as CNSs regions of the genome that could recently have been coding, had low alignment confidence or may have been missed in the gene annotation, the conserved regions were masked for the following: CDSs in the reference genome and CDSs lifted over (liftOver, Kent genome browser utilities, https://github.com/ucscGenomeBrowser/kent) from the other diploid annotations to the reference using the best orthologous chain files previously generated, regions outside of gene models with strong evidence for expression (> = 20 reads) from libraries SRR11216565 SRR11216566 SRR11216567 SRR11216568 SRR11216569 SRR11216570 SRR11216571 SRR11216573 SRR11216574 aligned to the *O.macrolepis* reference with bwa in mem mode v.0.7.13[89] and aggregated into a bedGraph with genomeCoverageBed (bedtools2, v.2.25.0)[103], regions annotated as TEs in the reference, regions with fewer than 9 alignments in the 12-way diploid multiple alignment.

For the tetraploid genomes, the two highest scoring chains for each region of the diploid reference were retained as best orthologous chain files mapping to the polyploid subgenomes. These were use to liftOver the CNS locations in the diploid reference to their two corresponding locations in the tetraploids with options -minMatch=0.5 -multiple -fudgeThick -minBlocks=0.01. The sequence for each CNS location in the tetraploids was generated from the resulting bed file and bedtoolsgetfasta[103]. A custom script (based on https://en.wikipedia.org/wiki/Edit_distance#Computation) was used to calculate the edit distance between the reference CNS and the CNS location in the tetraploid. In order to avoid short and likely noisy CNSs as well as regions that lifted over poorly, only CNSs of length between 8 and 100nt were considered. The minimum of the edit distance between the reference CNS and the tetraploid sequence and the reverse complemented reference CNS and the tetraploid sequence was divided by the maximum of the original CNS length and the lifted over sequence length to develop the normalized edit distance metric for CNSs. In order to generate a similar set of neutral sequences, the two tetraploid best orthologous chain files were converted to maf format as described above and msa-view was used to extract the 4D sites from both the reference and tetraploid subgenomes and randomize their order (for each file homeo0 and homeo1: msa_view species.homeo0.1.A.maf -o FASTA –4d –features genes.gff > 4d.species.A.ss, msa_view –in-format SS –out-format FASTA –randomize 4d.species.A.ss > 4d.species.A.-fasta). From this file of randomly shuffled 4D sites, a random location was selected and the corresponding sequences in the reference and tetraploid subgenomes were extracted and the same process of estimating a normalized edit distance described above was performed. For each subgenome of each species a mean normalized edit distance was generated for the CNSs and similarly for the neutral sites. Across the pairs of CNS and neutral values for each subgenome of each species, a least-squares fit was generated to model the global divergence of CNSs across all 10 subgenomes (r = 0.96) as a function of neutral phylogenetic distance from the reference. The shortest distance from this model-line to each individual subgenome point was used to generate an estimate of constraint departure from the model expectation with the vector sum of variation over chromosomes for neutral and CNS points used to generate a 1 SD error estimate (Supplementary Table 20).

## Identification and analysis of A/B compartments

We firstly parsed the HiC data by HiC-Pro v.3.1.0[152] (Supplementary Tables 6–8) and selected the HiC ice-matrix with 100 kb resolution for subsequent analysis. To identify the A and B compartments, we adapted the PCA-based method[153]. First, using corrected (ICE) interaction matrices for each chromosome at 100 kb resolution. Next, calculating the Pearson correlation and covariance matrices of this matrix. Third, PCA dimensionality reduction analysis is performed on the correlation coefficient matrix, and on the first principal component PC1 axis, the chromatin region can be clearly divided into two parts, called A/B compartments. All of these processes were qualified using cworld v.1.01 software (GitHub-dekkerlab/cworld-dekker: perl cworld module and collection of utility/analysis scripts for C data (3 C, 4 C, 5 C, Hi-C). After identification, the subgenome A/B compartment distribution of each chromosome under 100 kb was obtained (Supplementary Figs. 48–50). Combining the gene annotation files of each chromosome and the transposon annotation files, we quantitatively counted the number of genes and the proportion of transposons contained in the A compartment and the B compartment (Supplementary Data 5) through the bedtools intersect command of the bedtools v.2.30.0[103]. Gene expression was analyzed as the average TPM of three to three to five replicates for all genes in A/B compartments for a given tissue type.

## Identification and analysis of TAD

The HiC ice-matrix after HiC-Pro v.3.1.0[152] at 40 kb resolution was selected for subsequent analysis, and we used hicFINDTAD v.3.7.2[71] and HiTAD v.0.4.4[72] to identify TAD, respectively. Using the results of hicFINDTAD v.3.7.2[71] combined with gene annotation files, the number of genes per TAD boundary was counted using bedtools intersect. In the HiTAD v.0.4.4[72] package, first, the DI is calculated for each bin to measure the upstream/downstream interaction bias, and then a candidate boundary list is predicted by using the Hidden Markov Model. Instead of using a fixed window size in the traditional DI calculation, HiTAD v.0.4.4[72] estimates a dynamic window size for

each bin, which greatly increases its sensitivity. Under the HiTAD v.0.4.4[72] results we counted the size of each TAD, we used the JCVI v.1.1.13[93] software to perform collinearity analysis on the subP and subM. If the proportion of overlapping co-linked genes within TAD in each collinear block exceeds 70% and the number of genes in the block is greater than five or more, this region is considered as a conserved TAD.

## Reporting summary

Further information on research design is available in the Nature Portfolio Reporting Summary linked to this article.

## Data availability

All raw data of the HiFi, transcriptome, HiC, genome survey, methylation data as well as the 21 assembled genomes generated in this study have been deposited in the Genome Sequence Archive[154] and the Genome Warehouse[155] in National Genomics Data Center (NGDC)[156] (https://ngdc.cncb.ac.cn/) with BioProject ID PRJCA012952. The assembly mitochondrial genomes have been deposited in the GenBase with accession number C_AA001619.1 to C_AA001639.1 (https://ngdc.cncb.ac.cn/genbase/review/469c71a21459) in NGDC[156]. All released other data used in this study was listed in Supplementary Table 24.

## Code availability

All software used in this study are described in detail in the Methods. Custom scripts used for Gene and TE metaplots can be found in the GitHub using the following web site: https://github.com/niederhuth/methylation_scripts. The custom script used to calculate the edit distance between the reference CNS and the CNS location in the tetraploid can be found on the following website: https://en.wikipedia.org/wiki/Edit_distance#Computation.

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

## Acknowledgements

This work was supported by the National Natural Science Foundation of China (32260153, 31960035, 31700318 and 31560308 to H.H.Z. and 32272940 to M.J.H.), the Funds for Distinguished Young Scientists of Jiangxi Province (20192BCBL23028 to H.H.Z.), Key programs of Jiangxi Youth Science Foundation (20202ACBL215008 to H.H.Z.), Jiangxi fishery seed industry joint breeding project (2022yyzygg-04 to X.G.Z.), National Science Foundation Postdoctoral Research Fellowship in Biology (PRFB-2109178 to J.R.B.), and National Science Foundation (NSF-PGRP 2029959 to P.P.E.).

## Author contributions

H.Z. conceived the project. H.H.Z., M.J.H, P.P.E., M.S. and X.G.Z. designed and supervised this project. H.H.Z., M.R.X.X., X.G.Z., H.M.L., J.Y.Y., S.F.D. and J.G.X. collected fish samples for genome, transcriptome, HiC, and methylation sequencing. H.H.Z. and M.R.X.X. coordinated all sample sequencing in collaboration with Glbizzia Biosciences. M.R.X.X., Z.Y.L., G.Y.L., Z.Q.C., F.S., G.H.Y., S.P., M.J.H. and H.H.Z. performed genome assembly and annotation. M.R.X.X., G.Y.L. and S.P. identified Hox genes. Z.Y.L. determined subgenome-resolved genomes. Z.Y.L., K.D., J.V., K.H.W. and M.J.H. conducted the analyses of phylogenies and age and subgenome-biased index for TEs. K.D. performed the identification of orthologous genes and estimated the divergence time of species or subgenomes. M.R.X.X. and Z.N.G. performed evolutionary mechanisms of duplicated ohnologs. J.R.B, Y.H.W., A.G., A.E.P., S.J.T., K.B., C.E.N., P.P.E. M.J.H. and H.H.Z. generated subgenome dominance patterns in studied allotetraploids. H.H.Z. and P.P.E. wrote the article with input from co-authors. I.B., M.S. and A.N. revised the manuscript and participated in discussions.

## Competing interests

The authors declare no competing interests.

## Additional information

[1]College of Pharmacy and Life Science, Jiujiang University, Jiujiang, China. [2]Shenzhen Branch, Guangdong Laboratory for Lingnan Modern Agriculture, Genome Analysis Laboratory of the Ministry of Agriculture, Agricultural Genomics Institute at Shenzhen, Chinese Academy of Agricultural Sciences, Shenzhen, China. [3]Department of Horticulture, Michigan State University, East Lansing, MI, USA. [4]The Xiphophorus Genetic Stock Center, Texas State University, San Marcos, TX, USA. [5]College of Life Science and Agronomy, Zhoukou Normal University, Zhoukou, Henan, China. [6]Glbizzia Biosciences, Beijing, China. [7]Department of Plant Biology, Michigan State University, East Lansing, MI, USA. [8]Department of Integrative Biology, University of California Berkeley, Berkeley, CA, USA. [9]Department of Zoology, University of British Columbia, Vancouver, British Columbia, Canada. [10]Key Laboratory of Freshwater Fish Reproduction and Development (Ministry of Education), Southwest University, School of Life Sciences, Chongqing, China. [11]Department of Chromosome Biology, Max Planck Institute for Plant Breeding Research, Cologne, Germany. [12]Jiujiang Academy of Agricultural Sciences, Jiujiang, China. [13]Boyce Thompson Institute, Ithaca, NY, USA. [14]Department of Integrative Biology, Michigan State University, East Lansing, MI, USA. [15]Developmental Biochemistry, Biocenter, University of Würzburg, Würzburg, Bayern, Germany. [16]State Key Laboratory of Resource Insects, Key Laboratory for Sericulture Functional Genomics and Biotechnology of Agricultural Ministry, Southwest University, Chongqing, China. [17]These authors contributed equally: Min-Rui-Xuan Xu, Zhen-Yang Liao, Jordan R. Brock, Kang Du, Guo-Yin Li, Zhi-Qiang Chen. [18]These authors jointly supervised this work: Patrick P. Edger, Hua-Hao Zhang.
✉e-mail: zcj7820@163.com; phch1@biozentrum.uni-wuerzburg.de; pedger@gmail.com; minjinhan@126.com; zhanghuahao_0824@126.com

