## [Peer Review File · Nature Communications]

REVIEWER COMMENTS

Reviewer #1 (Remarks to the Author):

Xu et al. present high-quality genome assemblies of twenty-one cyprinid fishes (most contig level; some scaffolded with Hi-C data), and use these assemblies to test hypotheses regarding polyploidy and genome evolution following whole genome duplication (WGD). They use kmer analysis and other approaches to phase the subgenomes (assign chromosomes to their respective subgenomes), which allows for downstream analyses of subgenome dominance (SD). The evidence provided for WGDs in this paper helps contextualize the extensively studied carp/goldfish duplication that has been the focus of several high-profile papers in recent years. Understanding these duplications is key for illuminating the evolutionary history of this most (or nearly most) speciose group of vertebrates on the planet – the Cyprinidae – which also contains some of the most economically-important species for freshwater aquaculture (i.e., common carp, *Cyprinus carpio*; and grass carp, *Ctenopharyngodon idella*).

This is an impressive and important body of work. The twenty-one genome assemblies are high quality, despite lacking HiC or other scaffolding for most of them. In addition to the genome assemblies, tissue-specific RNA-seq data and DNA methylation calls are also presented. The analyses generally reflect state-of-the-art approaches and conclusions reached are well supported. The inclusion of TAD analysis from Hi-C data is an exciting addition (although the specifics are beyond my area of expertise to evaluate claims made). In general, the figures are mostly sufficient. I have some specific suggestions regarding figure quality below.

The work generally supports the conclusions and claims made. Aside from the phylogenetic analysis and inferred timing of diversification, the overall results and conclusions of the paper are mostly as expected and match previous studies cited in the paper. For example, (1) The karyotype of cyprinids is largely conserved, other than WGD events, with multiples of $n=25$ chromosomes typically present. (2) The 4R allopolyploidy events have been demonstrated previously in several other studies, including goldfish and common carp papers. (3) The maternal subgenome dominance and differences in TE profiles have also been shown previously in other cyprinid-focused papers. The main strength of the current study is that it greatly expands the taxonomic sampling beyond the common carp/goldfish duplication, which is a noteworthy and exciting contribution for showing the extent to which carp/goldfish patterns are generalizable. Overall, I think this is an important paper that will generate significant interest in the field of polyploidy and genome evolution.

Genome assemblies: The genomes are high quality, with all genomes having a contig N50 >4 mb, and most are around ~20mb contig N50. It would have been nice to see Hi-C data for the contig level assemblies, but I understand the need to move forward without it given the large number of genomes involved. Because synteny is so strongly conserved in cyprinids, it should still be possible to use the chromosome level assemblies as pseudoscaffolds for the contig level assemblies. I would feel more comfortable with orthology information based on pseudosynteny or microsynteny within contigs instead of just blastp results (see phylogenetics section below).

Phylogenetic analysis and dating: My main concerns about the manuscript rest with the phylogenetics portion of the study. The use of a single outgroup (Triplophysa, a loach in the Cobitoidea) is worrisome, particularly since the sister group to Cyprinidae is a contentious issue and significant gene-tree discordance exists, perhaps because of frequent introgressive hybridization in both loaches and cyprinids. The main time calibrations used in this study stem from a paper by Li and Guo (2020; ref 124), which used the earliest definite cyprinid fossils as an age constraint on the origins of Cyprinidae; however, the conclusions of that paper are quite different from almost every other cyprinid time-calibrated phylogeny (see timetree.org for a summary of timeline estimates for Cyprinidae). In addition to calibration issues, I am concerned that any errors in orthology/phasing lead to biases in inferred timelines when using concatenation methods. Gene-trees were also used, but the amount of gene-tree discordance should be presented more clearly, particularly since the evolutionary history of cypriniform fishes is rich with reticulate evolution via introgressive hybridization (and not only in allopolyploids). How this evolutionary complexity affects inferred dates is an important question that should be considered

here.

Subgenome phasing: It was explicitly noted in the text that subgenome designations do not denote common ancestry (i.e., subA in one species is not necessarily the same as subA in another; lines 230-231). Previous studies used 'M' and 'P' to denote the maternal and paternal subgenomes throughout, which makes it easier to track these through the figures and manuscript (e.g., Luo et al. 2020); is it possible to do so here, instead of the subA and subB designations? I understand that some of these are also independently derived, but it would make the text easier to follow and better connect the title of the paper to the specific results.

Methodological details: Overall, the bioinformatic analyses are appropriate and represent state-of-the-art in the field; however, the details are under reported and the work is not reproducible as presented. More analytical detail could mitigate this omission (and make the results more broadly useful for other systems).

Minor comments

Line 136-145. The wording here implies that this three-gene study is the only work that has been done on cyprinine phylogenetics. Most of the recent carp and goldfish genome papers have included phylogenetic trees based on hundreds or thousands of genes from the available genomes across the entire Cyprinidae (more seem to be included in each subsequent paper) and other, more focused systematics work has been conducted on this taxon. This section should be revised to better reflect the literature.

Line 170 – What is the citation for the chromosome information? I don't think Yang et al did any karyotyping, but rather pasted results from other sources?

Line 300-301 – Six tissues were examined for expression levels. The low level of sub-functionalization inferred (<1% of gene pairs) could be due to the relatively small number of tissues examined? Presumably the proportion of genes with tissue-specific expression in at least one tissue should increase as additional tissues are examined, particularly as different ontogenetic stages were sampled (see, e.g., Li et al. 2021 Nature Genetics; Luo et al. 2020 Science Advances).

Line 727-729 – This strict synteny approach seems more likely to be useful for identifying proper orthology assignments than the blastp based approach, especially given the mostly conserved synteny in cyprinids.

Line 764-766 – the time calibrations used here are poorly (or not) justified. Two of the three calibration points lack evidence; one is based on an outlier study (as noted above). It would be better to base calibration points on the fossil record or definitive biogeographic events.

Fig 1 – Triplophysa vs cyprinids; 55 ma divergence time seems like a significant underestimate (97 ma per timetree.org). There also appears to be some taxon sampling gaps here; there are several *Sinocyclocheilus* and *Labeo* genomes that would be helpful to the placement of the WGD events. It would also be useful to see the inferred timing of WGD mapped onto this tree (circles represent the speciation events between diploid progenitors as noted, but not the allopolyploidy event itself, as noted).

Fig 2b – What is the green band on chr 11? Apologies if I have missed this in the text somewhere.

Fig 3. I understand the goal of this figure, but it is a little hard to follow. Perhaps instead of the dotted lines the dates could be added in parentheses after Ks peak values (color coded), and subsequently listed on the tree?

Supp Fig 2a,d – What are the inferred chromosome boundaries on the HiC heatmap (panel A)? Also, I cannot read the TE type names in panel D.

Supp Fig 12,13 – As noted above, the divergence time between the outgroup (Triplophysa) and the ingroup (Cyprinidae) is <60 million years here; this is much more recent than has been estimated in other studies. For example, timetree.org lists the divergence at 97 million years (81-

109 ma confidence interval). There is no measure of node date uncertainty on this tree. Part of the issue here could be because of the possible hybridization history of loaches (Triphopsya) and their uncertain phylogenetic position relative to cyprinids, catostomids, and gyrinocheilids, making this a poor (solo) outgroup for the family. It would be better to use multiple outgroups, and add confidence intervals for node-date uncertainty to this tree. I am skeptical about concatenation methods and the use of single copy orthologs, especially since the latter are disproportionately retained on one subgenome (and therefore are likely to give different dates than non-biased genes – see previous work on the carp/goldfish duplication for examples of single copy BUSCO subgenome retention bias).

Supp Table 2. A total of 20 species are present here, with *S. cirriculus* missing. It would be helpful to add full genera names to this table (and supp. tables 3-6), particularly since the abbreviations correspond to multiple genera (e.g., *S. acanthopterus* and *S. procheilus* are different genera).

Supp Table 3. I am confused by 'Inset' size of 350 and 15000 bp. Does this mean target fragment size, or perhaps target insert size?

Supp Table 4. Heterozygous rate listed in two columns. There are other examples of similar errors in the supplement that need careful review.

Supp Table 5 is missing?

Percent duplicated BUSCO genes – the high percentage of duplicated BUSCO genes (>>50%) is perhaps a function of the annotation methods employed; in particular, we have seen an overestimation when using GeMoMa (one of three methods employed). The percentage of duplicates noted here is significantly higher than other cyprinid allopolyploidy events.

Reviewer #2 (Remarks to the Author):

Review of Xu et al., "Maternal dominance contributes to subgenome differentiation in allopolyploid fishes."

Overview: In this manuscript, the authors obtain genome sequences and other genomic data for 21 fishes that are relatives of goldfish to better understand the patterns of polyploidy among these fishes. They present evidence for 3 different allopolyploidy events among the species in question and show very interesting evidence that the expression and gene loss biases seen for these polyploidies are driven by over-expression of the maternal subgenome in the allopolyploid relative to the paternal.

Major comments:

This is a very exciting paper that considerably enhances our understanding of why allopolyploid genomes show biases in expression and loss patterns. The data collected are extremely extensive and impressive. I do have a few suggestions and comments:

Lines 228-229: This sentence is a bit unclear—I think what is meant is that subA and subB does not have a consist between different genomes, but a little work could help.

Lines 298-299: While I understand the goal of this analysis, I think this approach of classifying retained duplicates by their mechanism of evolutionary retention is oversimplified. While I think it is ok to leave the section in the manuscript, it should be qualified with some discussion of the fact that a mechanism of retention is not strictly inferable for duplicated genes. For instance, see (Conant, Birchler, and Pires 2014).

Line 397 and following: This section seems rather long and (to my mind) reports an ambiguous result. Perhaps it could be shortened or omitted.

Discussion: I think the major omission in the Discussion is a lack of a model or hypothesis for why maternal subgenomes show dominance. I am aware that this might be speculative, but given the

striking nature of the results, I think it is worth commenting on.

Finally, I think it is worth commenting on the potential problems of reciprocal gene loss (RGLs) in the phylogenic pipeline. The ortholog calling in the larger analysis uses only sequence similarity, which could easily mistake 1:1 RGLs for orthologs. Again, I don't think the analyses need be changed, but the authors should mention the potential for such difficulties.

Minor comments:

There also a fair number of typos here and there:

Line 205: "assemblies." See also line 602&875m for instance.

Reference:

Conant, G. C., J. A. Birchler, and J. C. Pires. 2014. Dosage, duplication, and diploidization: clarifying the interplay of multiple models for duplicate gene evolution over time. *Current Opinion in Plant Biology* 19:91-98.

Reviewer #3 (Remarks to the Author):

By generating sequencing data of genomes, transcriptomes, and epigenomes, Zhang and colleagues tried to characterize subgenome differentiation in allopolyploid fishes. In general, the study would be of interest to those who are working on fish and polyploid evolution. However, there are scientific weaknesses. One of the most concern is that all so-called novel findings have been shown in earlier studies either in common carps and goldfishes or plants. For example, the maternal dominance contributes to subgenome differentiation have been found in polyploid plants since studies with qPCR-based expression studies; the asymmetric divergence between subgenomes in cyprinids has been shown in common carps; the correlation between subgenome dominance and transposable element densities has no hard evidence but speculation. Therefore, the study is like a collection of polyploid genomic study by including many aspects, but no finding is novel to our understanding of evolutionary genomics in polyploids.

RESPONSE TO REVIEWERS' COMMENTS

Reviewer #1 (Remarks to the Author):

Xu et al. present high-quality genome assemblies of twenty-one cyprinid fishes (most contig level; some scaffolded with Hi-C data), and use these assemblies to test hypotheses regarding polyploidy and genome evolution following whole genome duplication (WGD). They use kmer analysis and other approaches to phase the subgenomes (assign chromosomes to their respective subgenomes), which allows for downstream analyses of subgenome dominance (SD). The evidence provided for WGDs in this paper helps contextualize the extensively studied carp/goldfish duplication that has been the focus of several high-profile papers in recent years. Understanding these duplications is key for illuminating the evolutionary history of this most (or nearly most) speciose group of vertebrates on the planet – the Cyprinidae – which also contains some of the most economically-important species for freshwater aquaculture (i.e., common carp, *Cyprinus carpio*; and grass carp, *Ctenopharyngodon idella*).

This is an impressive and important body of work. The twenty-one genome assemblies are high quality, despite lacking HiC or other scaffolding for most of them. In addition to the genome assemblies, tissue-specific RNA-seq data and DNA methylation calls are also presented. The analyses generally reflect state-of-the-art approaches and conclusions reached are well supported. The inclusion of TAD analysis from Hi-C data is an exciting addition (although the specifics are beyond my area of expertise to evaluate claims made). In general, the figures are mostly sufficient. I have some specific suggestions regarding figure quality below.

The work generally supports the conclusions and claims made. Aside from the phylogenetic analysis and inferred timing of diversification, the overall results and conclusions of the paper are mostly as expected and match previous studies cited in the paper. For example, (1) The karyotype of cyprinids is largely conserved, other than WGD events, with multiples of $n=25$ chromosomes typically present. (2) The 4R allopolyploidy events have been demonstrated previously in several other studies, including goldfish and common carp papers. (3) The maternal subgenome dominance and differences in TE profiles have also been shown previously in other cyprinid-focused papers. The main strength of the current study is that it greatly expands the taxonomic sampling beyond the common carp/goldfish duplication, which is a noteworthy and exciting contribution for showing the extent to which carp/goldfish patterns are generalizable. Overall, I think this is an important paper that will generate significant interest in the field of polyploidy and genome evolution.

Genome assemblies: The genomes are high quality, with all genomes having a contig N50 >4 mb, and most are around ~20mb contig N50. It would have been nice to see Hi-C data for the contig level assemblies, but I understand the need to move forward without it given the large number of genomes involved. Because synteny is so strongly conserved in cyprinids, it should still be possible to use the chromosome level assemblies as pseudoscaffolds for the contig level assemblies. I would feel more comfortable with orthology information based on pseudosynteny or microsynteny within contigs instead of just blastp results (see phylogenetics section below).

*Thanks! Besides the blastp approach, we also implemented a strict synteny confirmation where at least five orthologs have to be arranged in a row with the largest gap being fewer than five genes. Finally, 310 gene orthologs with a 1:2 relationship were obtained. We found that the topology structure (Supplementary Figure 14) of the phylogeny using zebrafish as the outgroup was almost consistent with that of the origin phylogeny (Figure 1), which used *Triplophysa bleekeri* as the outgroup. Please also see the below response for detail.*

Phylogenetic analysis and dating: My main concerns about the manuscript rest with the phylogenetics portion of the study. The use of a single outgroup (*Triplophysa*, a loach in the Cobitoidea) is worrisome, particularly since the sister group to Cyprinidae is a contentious issue and significant gene-tree discordance exists, perhaps because of frequent introgressive hybridization in both loaches and cyprinids. The main time calibrations used in this study stem from a paper by Li and Guo (2020; ref 124), which used the earliest definite cyprinid fossils as an age constraint on the origins of Cyprinidae; however, the conclusions of that paper are quite different from almost every other cyprinid time-calibrated phylogeny (see timetree.org for a summary of timeline estimates for Cyprinidae). In addition to calibration issues, I am concerned that any errors in orthology/phasing lead to biases in inferred timelines when using concatenation methods. Gene-trees were also used, but the amount of gene-tree discordance should be presented more clearly, particularly since the evolutionary history of cypriniform fishes is rich with reticulate evolution via introgressive hybridization (and not only in allopolyploids). How this evolutionary complexity affects inferred dates is an important question that should be considered here.

*I fully agreed this point. To estimate the effect of selected outgroup, we have rebuilt the phylogeny using *Danio rerio* and *Danionella translucida* as the outgroup. To identify gene orthologs with a 1:2 relationship (one copy in diploids, and one copy on the subP and one copy on the subM) in these species, besides the blastp approach, we also implemented a strict synteny confirmation where at least five orthologs have to be arranged in a row with the largest gap being fewer than five genes. Finally, 310 gene orthologs with a 1:2 relationship were obtained. We found that the topology structure (Supplementary Figure 14) was almost consistent with that of the origin phylogeny (Figure 1), which used *Triplophysa bleekeri* as the outgroup. Therefore, we think the outgroup might not have great effect on our results. Therefore, we keep the origin phylogeny of Figure 1 in the main text, and the rebuilt phylogeny in Supplementary information.*

Besides the concatenation methods, we rebuilt the consensus species tree and individual gene trees using these 1,669 single-copy orthologs (the above similar method was used to identify gene orthologs with a 1:2 relationship) to further determine the relationship of the maternal and paternal diploid progenitors of known polyploids with possible diploids. It showed that the consensus species tree was indeed consistent with the concatenation-base analyses (Supplementary Figures 18 and 19). However, we also observed the differences between overall consensus species tree and individual gene trees, which was normally as a result of incomplete lineage sorting or ongoing gene flow between the species. We have discussed this point in the revised manuscript.

*To estimate the divergence time of all species, four time calibrations from timetree and a recent paper (Feng et al., 2023. Monsoon boosted radiation of the endemic East Asian carps. *Sci China Life Sci.* 2023 Mar;66(3):563-578.) were set: 81.9-100.1 Ma to the most recent common ancestor of *Nemacheilidae* and *Cyprinidae*; 27.82-33.9 Ma to the most recent common ancestor of*

Barbinae and Cyprininae; 40.4-48.6 Ma to the root of Cyprinidae; and 11.6-12.7 Ma to crown age of Leuciscinae. It showed that Triplophysa and cyprinids diverged about 90Ma (Figure 1). Meanwhile, the divergence time of all subfamilies was similar to those reported by Feng et al., 2023.

Subgenome phasing: It was explicitly noted in the text that subgenome designations do not denote common ancestry (i.e., subA in one species is not necessarily the same as subA in another; lines 230-231). Previous studies used ‘M’ and ‘P’ to denote the maternal and paternal subgenomes throughout, which makes it easier to track these through the figures and manuscript (e.g., Luo et al. 2020); is it possible to do so here, instead of the subA and subB designations? I understand that some of these are also independently derived, but it would make the text easier to follow and better connect the title of the paper to the specific results.

I fully agreed this point. We have used the subP and subM designations in the whole manuscript and figures.

Methodological details: Overall, the bioinformatic analyses are appropriate and represent state-of-the-art in the field; however, the details are under reported and the work is not reproducible as presented. More analytical detail could mitigate this omission (and make the results more broadly useful for other systems).

Thanks! We have made some more details in the Method part.

Minor comments

Line 136-145. The wording here implies that this three-gene study is the only work that has been done on cyprinine phylogenetics. Most of the recent carp and goldfish genome papers have included phylogenetic trees based on hundreds or thousands of genes from the available genomes across the entire Cyprinidae (more seem to be included in each subsequent paper) and other, more focused systematics work has been conducted on this taxon. This section should be revised to better reflect the literature.

Thanks! You know that the major results in our manuscript enhance the underlying genetic mechanisms contributing to subgenome dominance in allopolyploid fishes. To address this point, we should firstly determine the maternal and paternal diploid progenitors of these polyploids. To our best knowledge, Yang et al., 2022 is the only work that tried to address this point within this group, but the authors in this paper only used three single-copy nuclear loci. However, the phylogenetic history of these three genes may not reflect the true history of species relationships within this subfamily. We have revised this section to better reflect the literature ().

Line 170 – What is the citation for the chromosome information? I don’t think Yang et al did any karyotyping, but rather pasted results from other sources?

Thanks! We had added three papers (in Chinese) related to the karyotype analysis of three species (Procypris rabaudi, Spinibarbus sinensis and Luciobarbus capito) in the Reference, and it showed that three species included 100 chromosomes.

Line 300-301 – Six tissues were examined for expression levels. The low level of sub-functionalization inferred (<1% of gene pairs) could be due to the relatively small number of

tissues examined? Presumably the proportion of genes with tissue-specific expression in at least one tissue should increase as additional tissues are examined, particularly as different ontogenetic stages were sampled (see, e.g., Li et al. 2021 Nature Genetics; Luo et al. 2020 Science Advances).

Thanks! We have discussed this at the end of this section.

Line 727-729 – This strict synteny approach seems more likely to be useful for identifying proper orthology assignments than the blastp based approach, especially given the mostly conserved synteny in cyprinids.

*Thanks! We have also determined single-copy orthologs of these species using the blastp based approach and implemented a strict synteny confirmation where at least five orthologs have to be arranged in a row with the largest gap being fewer than five genes. Finally, we obtained 310 single-copy orthologs. Then, we rebuilt the phylogeny using *Danio rerio* and *Danionella translucida* as the outgroup (Supplementary Figure 14). We found that the topology structure was almost consistent with that of the origin phylogeny (Figure 1), which used *Triplophysa bleekeri* as the outgroup.*

Line 764-766 – the time calibrations used here are poorly (or not) justified. Two of the three calibration points lack evidence; one is based on an outlier study (as noted above). It would be better to base calibration points on the fossil record or definitive biogeographic events.

*Thanks! To estimate the divergence time of all species, four time calibrations from timetree and a recent paper (Feng et al., 2023. Monsoon boosted radiation of the endemic East Asian carps. *Sci China Life Sci.* 2023 Mar;66(3):563-578.) were set: 81.9-100.1 Ma to the most recent common ancestor of *Nemacheilidae* and *Cyprinidae*; 27.82-33.9 Ma to the most recent common ancestor of *Barbinae* and *Cyprininae*; 40.4-48.6 Ma to the root of *Cyprinidae*; and 11.6-12.7 Ma to crown age of *Leuciscinae*. It showed that *Triplophysa* and cyprinids diverged about 90Ma (Figure 1 and Supplementary Figure 12). Meanwhile, the divergence time of all subfamilies was similar to those reported by Feng et al., 2023.*

Fig 1 – *Triplophysa* vs cyprinids; 55 ma divergence time seems like a significant underestimate (97 ma per timetree.org). There also appears to be some taxon sampling gaps here; there are several *Sinocyclocheilus* and *Labeo* genomes that would be helpful to the placement of the WGD events. It would also be useful to see the inferred timing of WGD mapped onto this tree (circles represent the speciation events between diploid progenitors as noted, but not the allopolyploidy event itself, as noted).

*Thanks! We have re-estimated the divergence time of *Triplophysa* vs cyprinids using a time calibration from timetree, and It showed that *Triplophysa* and cyprinids diverged about 90Ma (Figure 1 and Supplementary Figure 12). For *Sinocyclocheilus*, there is no available assembly with subgenome-resolved, which make us difficult to use it in this manuscript to determine the maternal and paternal diploid progenitors of these polyploids. For *Labeo*, we have sequenced three new species (*Crossocheilus oblongus*, *Semilabeo prochilus* and *Sinilabeo rendahli*) belonging to this lineage, and all of them had been used in the phylogeny. Yes, we had added the inferred timing of WGD mapped onto the revised Figure 1.*

Fig 2b – What is the green band on chr 11? Apologies if I have missed this in the text somewhere.

Thanks! The green band showed one example of a collinearity gene between homologous chromosomes. We have made this point clearly in the figure legends.

Fig 3. I understand the goal of this figure, but it is a little hard to follow. Perhaps instead of the dotted lines the dates could be added in parentheses after Ks peak values (color coded), and subsequently listed on the tree?

Thanks! We have revised this figure according to your suggestion.

Supp Fig 2a,d – What are the inferred chromosome boundaries on the HiC heatmap (panel A)? Also, I cannot read the TE type names in panel D.

Thanks! The chromosome boundaries were inferred according to contact signals of the Hi-C data for all chromosomes of each genome. The TE type names were also showed in panel D.

Supp Fig 12,13 – As noted above, the divergence time between the outgroup (Triplophysa) and the ingroup (Cyprinidae) is <60 million years here; this is much more recent than has been estimated in other studies. For example, timetree.org lists the divergence at 97 million years (81-109 ma confidence interval). There is no measure of node date uncertainty on this tree. Part of the issue here could be because of the possible hybridization history of loaches (Triplophysa) and their uncertain phylogenetic position relative to cyprinids, catostomids, and gyrinocheilids, making this a poor (solo) outgroup for the family. It would be better to use multiple outgroups, and add confidence intervals for node-date uncertainty to this tree. I am skeptical about concatenation methods and the use of single copy orthologs, especially since the latter are disproportionately retained on one subgenome (and therefore are likely to give different dates than non-biased genes – see previous work on the carp/goldfish duplication for examples of single copy BUSCO subgenome retention bias).

Thanks! We obtained gene orthologs with a 1:2 relationship (one copy in diploids, and one copy on the subP and one copy on the subM) for the phylogenies. We have made this point clearly in the revised manuscript. We also re-estimated the divergence time of all species using four time calibrations from timetree and a recent paper (Feng et al., 2023. Monsoon boosted radiation of the endemic East Asian carps. Sci China Life Sci. 2023 Mar;66(3):563-578.). Meanwhile, we rebuilt the phylogeny using zebrafish as the outgroup. Please see more details in the above response.

Supp Table 2. A total of 20 species are present here, with *S. cirriculus* missing. It would be helpful to add full genera names to this table (and supp. tables 3-6), particularly since the abbreviations correspond to multiple genera (e.g., *S. acanthopterus* and *S. procheilus* are different genera).

Thanks! We have revised this in these supp. Tables. If two species share the same first letter from two different genera, we used the first two letters for one species to distinguish from the other.

Supp Table 3. I am confused by ‘Inset’ size of 350 and 15000 bp. Does this mean target fragment size, or perhaps target insert size?

Thanks! ‘Inset’ size of 350 and 15000 bp meant target fragment size. I am sorry for a typo of insert. We have revised this.

Supp Table 4. Heterozygous rate listed in two columns. There are other examples of similar errors in the supplement that need careful review.

Thanks! We have carefully checked this.

Supp Table 5 is missing?

Thanks! We have added this table in the revised manuscript.

Percent duplicated BUSCO genes – the high percentage of duplicated BUSCO genes (>>50%) is perhaps a function of the annotation methods employed; in particular, we have seen an overestimation when using GeMoMa (one of three methods employed). The percentage of duplicates noted here is significantly higher than other cyprinid allopolyploidy events.

Thanks! For gene annotation, besides Illumina data, we also used Iso-Seq data. For other reported cyprinid allopolyploidy, they only used Illumina data for gene annotation. Therefore, we think it may make some progress in this study combining Illumina data and Iso-Seq data for gene annotation. This method may obtain more duplicated genes in our sequenced species.

Reviewer #2 (Remarks to the Author):

Review of Xu et al., “Maternal dominance contributes to subgenome differentiation in allopolyploid fishes.”

Overview: In this manuscript, the authors obtain genome sequences and other genomic data for 21 fishes that are relatives of goldfish to better understand the patterns of polyploidy among these fishes. They present evidence for 3 different allopolyploidy events among the species in question and show very interesting evidence that the expression and gene loss biases seen for these polyploidies are driven by over-expression of the maternal subgenome in the allopolyploid relative to the paternal.

Major comments:

This is a very exciting paper that considerably enhances our understanding of why allopolyploid genomes show biases in expression and loss patterns. The data collected are extremely extensive and impressive. I do have a few suggestions and comments:

Lines 228-229: This sentence is a bit unclear—I think what is meant is that subA and subB does not have a consist between different genomes, but a little work could help.

Thanks! Previous studies used ‘M’ and ‘P’ to denote the maternal and paternal subgenomes. Thus, the subP and subM denotes the paternal and maternal subgenome in the revised manuscript.

Lines 298-299: While I understand the goal of this analysis, I think this approach of classifying retained duplicates by their mechanism of evolutionary retention is oversimplified. While I think it is ok to leave the section in the manuscript, it should be qualified with some discussion of the fact that a mechanism of retention is not strictly inferable for duplicated genes. For instance, see (Conant, Birchler, and Pires 2014).

Thanks! We have made some discussion on this point in the revised manuscript.

Line 397 and following: This section seems rather long and (to my mind) reports an ambiguous result. Perhaps it could be shortened or omitted.

Thanks! We have shortened this section.

Discussion: I think the major omission in the Discussion is a lack of a model or hypothesis for why maternal subgenomes show dominance. I am aware that this might be speculative, but given the striking nature of the results, I think it is worth commenting on.

Thanks! We had added a model for explanation of our observed results in the Discussion as follows: The mitochondrial proteome contains products from over a thousand genes, while the mitochondrial genome encodes approximately only 13 proteins (i.e. 1% of the proteome). The vast majority of genes are now nuclear genome encoded following the horizontal gene transfer from the organellar genome to the nuclear genome over the past hundred million years. However, these nuclear genes might encode dosage-sensitive proteins that function in either organellar signaling networks or macromolecular complexes that must maintain proper stoichiometric balance with interacting partner (s) that are encoded in the organellar genome. Furthermore, the sequence of

the proteins encoded by both organellar and nuclear-encoded mitochondrial genes may have diverged among the diploid progenitors. Thus, there ' s a possibility for incompatibilities to arise from "mismatches" between the genes contributed by the paternal subgenome and the organellar genomes contributed by the maternal parents in allopolyploids. The biased expression of the maternal nuclear copy would resolve any potential conflicts. The model that we are proposing here is that observed dominance patterns in these allopolyploids is to preserve proper cytonuclear interactions, and ultimately, core cellular functions.

Finally, I think it is worth commenting on the potential problems of reciprocal gene loss (RGLs) in the phylogenetic pipeline. The ortholog calling in the larger analysis uses only sequence similarity, which could easily mistake 1:1 RGLs for orthologs. Again, I don't think the analyses need be changed, but the authors should mention the potential for such difficulties.

*Thanks! Besides the blastp approach, we also implemented a strict synteny confirmation where at least five orthologs have to be arranged in a row with the largest gap being fewer than five genes. We had made this point clearly in the revised manuscript. Finally, 310 gene orthologs with a 1:2 relationship (one copy in diploids, and one copy on the subP and one copy on the subM) were obtained. We found that the topology structure (Supplementary Figure 14) of the phylogeny using zebrafish as the outgroup was almost consistent with that of the origin phylogeny (Figure 1), which used *Triplophysa bleekeri* as the outgroup. Therefore, we keep the origin phylogeny of Figure 1 in the main text, and the rebuilt phylogeny in Supplementary Figure 14.*

Minor comments:

There also a fair number of typos here and there:

Line 205: "assemblies." See also line 602&875m for instance.

Thanks! We have checked this across the whole manuscript.

Reference:

Conant, G. C., J. A. Birchler, and J. C. Pires. 2014. Dosage, duplication, and diploidization: clarifying the interplay of multiple models for duplicate gene evolution over time. *Current Opinion in Plant Biology* 19:91-98.

Reviewer #3 (Remarks to the Author):

By generating sequencing data of genomes, transcriptomes, and epigenomes, Zhang and colleagues tried to characterize subgenome differentiation in allopolyploid fishes. In general, the study would be of interest to those who are working on fish and polyploid evolution. However, there are scientific weaknesses. One of the most concern is that all so-called novel findings have been shown in earlier studies either in common carps and goldfishes or plants. For example, the maternal dominance contributes to subgenome differentiation have been found in polyploid plants since studies with qPCR-based expression studies; the asymmetric divergence between subgenomes in cyprinids has been shown in common carps; the correlation between subgenome dominance and transposable element densities has no hard evidence but speculation. Therefore, the study is like a collection of polyploid genomic study by including many aspects, but no finding is novel to our understanding of evolutionary genomics in polyploids.

Thank you for taking the time to review our manuscript. If a consequence of polyploidy has been found in plants (which I agree there has been a long history dating back to qPCR-based expression studies), there is no expectation that similar impacts will be found in animals. Their shared heritage is simply too distant to suggest that these developed in a common ancestor. So a similar finding is quite notable and the whole package here allows the direct comparison of polyploidy in very distant systems to be compared as a whole rather than in a piecemeal fashion. Major differences in subgenome dominance is also commonly reported among different polyploid plants.

In the present study, we resolved the phylogenetic relationships among several key Cyprininae species, uncovered the polyploid origin of three allopolyploid species, and identified the closest extant relatives of their diploid progenitors. Yes, we agree - previous studies reported evidence for subgenome dominance among cyprinid fishes (which has emerged as a major model system for polyploidy in animals). However, as part of this study, we not only constructed a robust phylogenomic framework for the subfamily Cyprininae to phylogenetically localize polyploidy events but also to permitted us to investigate the underlying genetic mechanisms contributing to subgenome dominance in these allopolyploid fish. Furthermore, the maternal and paternal diploid progenitors of known polyploids in this group have remained unknown until now. To the best of our knowledge, this is the first study to show the potential role of maternal dominance in contributing to subgenome dominance in any allopolyploid animal. We have added new additional text in the discussion section as requested by the other reviewers. Plus, as noted in the manuscript, we cannot exclude the possibility that some of the observed subgenome expression differences, particularly at the individual gene level, is due to differences in DNA methylation and transposable element density differences as hypothesized in previous studies (largely from allopolyploid plants). Future studies of other allopolyploids are needed to determine if these observed patterns are shared by other allopolyploid fish/animals or are potentially unique to cyprinids.

In addition, we assembled de novo high-quality reference genomes of twenty-one cyprinid fishes and new large-scale genomic resources for the community as a foundation for future studies.

REVIEWERS' COMMENTS

Reviewer #1 (Remarks to the Author):

The authors have provided detailed responses to my previous concerns and this iteration of the manuscript is significantly improved from before. In particular, the revised time-calibrated phylogeny is much more consistent with the fossil record, geologic history, and our general understanding of cypriniform fish evolution. Assigning subgenomes to maternal and paternal origins also improves the strength of the findings over the previous version.

The developmental programs of fish and plants are very different. An important strength of the paper is broadening the phylogenetic scope of our understanding of mechanisms underlying subgenome dominance (or not) after WGD, especially since there are far fewer examples of allopolyploidy in animals than plants and the patterns of genome evolution seem at least somewhat different. This paper represents a step in that direction.

I suspect there will be significant pushback from the community about the conclusion of very low levels of subfunctionalization inferred (noting the caveat added on lines 301-302 in the current version). I certainly think this conclusion about dosage constrain is probably wrong and artifactual as noted in my previous review, but this is a relatively minor point in a much larger paper. The obvious solution would be to add more RNA-seq from different tissues and life stages, but I do not think it is worth holding the paper up for this addition.

Minor comments:

Supp. Fig 2b, 3b – it would help to keep order consistent across these – e.g., with subM on the top and subP on the bottom, and color schemes the same across figs.

Reviewer #2 (Remarks to the Author):

My previous comments have been addressed, and I have no further concerns.

RESPONSE TO REVIEWERS' COMMENTS

REVIEWER COMMENTS

Reviewer #1 (Remarks to the Author):

The authors have provided detailed responses to my previous concerns and this iteration of the manuscript is significantly improved from before. In particular, the revised time-calibrated phylogeny is much more consistent with the fossil record, geologic history, and our general understanding of cypriniform fish evolution. Assigning subgenomes to maternal and paternal origins also improves the strength of the findings over the previous version.

The developmental programs of fish and plants are very different. An important strength of the paper is broadening the phylogenetic scope of our understanding of mechanisms underlying subgenome dominance (or not) after WGD, especially since there are far fewer examples of allopolyploidy in animals than plants and the patterns of genome evolution seem at least somewhat different. This paper represents a step in that direction.

I suspect there will be significant pushback from the community about the conclusion of very low levels of subfunctionalization inferred (noting the caveat added on lines 301-302 in the current version). I certainly think this conclusion about dosage constrain is probably wrong and artifactual as noted in my previous review, but this is a relatively minor point in a much larger paper. The obvious solution would be to add more RNA-seq from different tissues and life stages, but I do not think it is worth holding the paper up for this addition.

Minor comments:

Supp. Fig 2b, 3b – it would help to keep order consistent across these – e.g., with subM on the top and subP on the bottom, and color schemes the same across figs.

Thanks! We have made this revision in the revised manuscript.

Reviewer #2 (Remarks to the Author):

My previous comments have been addressed, and I have no further concerns.

Thanks for your comments!